# Construction of a two-dimensional artificial antioxidase for nanocatalytic rheumatoid arthritis treatment

Bowen Yang [1,2], Heliang Yao[1], Jiacai Yang[1,2], Chang Chen[1,2] & Jianlin Shi [1,3 ✉]

Constructing nanomaterials mimicking the coordination environments of natural enzymes may achieve biomimetic catalysis. Here we construct a two-dimensional (2D) metal-organic framework (MOF) nanosheet catalyst as an artificial antioxidase for nanocatalytic rheumatoid arthritis treatment. The 2D MOF periodically assembles numbers of manganese porphyrin molecules, which has a metal coordination geometry analogous to those of two typical antioxidases, human mitochondrial manganese superoxide dismutase (Mn-SOD) and human erythrocyte catalase. The zinc atoms of the 2D MOF regulate the metal-centered redox potential of coordinated manganese porphyrin ligand, endowing the nanosheet with both SOD- and catalase-like activities. Cellular experiments show unique anti-inflammatory and pro-biomineralization performances of the 2D MOF, while in vivo animal model further demonstrates its desirable antiarthritic efficacy. It is expected that such a nanocatalytic antioxidation concept may provide feasible approaches to future anti-inflammatory treatments.

[1] State Key Laboratory of High Performance Ceramics and Superfine Microstructure, Shanghai Institute of Ceramics, Chinese Academy of Sciences; Research Unit of Nanocatalytic Medicine in Specific Therapy for Serious Disease, Chinese Academy of Medical Sciences (2021RU012), Shanghai 200050, P. R. China. [2] Center of Materials Science and Optoelectronics Engineering, University of Chinese Academy of Sciences, Beijing 100049, P. R. China. [3] Tenth People's Hospital and Shanghai Frontiers Science Center of Nanocatalytic Medicine, School of Medicine, Tongji University, Shanghai 200092, P. R. China. ✉email: jlshi@mail.sic.ac.cn

The coordination geometry of active metal sites of metalloenzymes underpins their in vivo catalytic kinetics and thermodynamics. For antioxidases, such as human mitochondrial manganese superoxide dismutase (Mn-SOD, SOD2) (Fig. 1a and Supplementary Fig. 1a), the active $Mn^{III}$ center responsible for superoxide anion ($O_2^{\bullet-}$) dismutation is coordinated with a solvent molecule ($OH^-$) and a histidine (His 27) in trans-axial positions, as well as two histidines (His 75 and His 164) and one aspartate (Asp 160) in the equatorial plane, forming a distorted pentacoordinated trigonal bipyramidal geometry[1]. Another typical paradigm is human erythrocyte catalase (Fig. 1b and Supplementary Fig. 1b), whose active $Fe^{III}$ center is located in the center of heme (Fe-protoporphyrin IX (Fe-PpIX)), while the axially-ligated tyrosine residue (Tyr 354) can regulate the metal-centered redox potential of heme favoring hydrogen peroxide ($H_2O_2$) decomposition[2]. Both enzymes show innate electrostatic facilitation mechanism that can guide the entrance of reactive oxygen species (ROS) into the metal sites for enabling redox reactions.

Given that numbers of inflammation-related diseases, such as rheumatoid arthritis, are associated with the downregulation of antioxidases[3,4], while the direct administration of these natural

**Fig. 1 Design concept of ZMTP nanosheets for catalytic rheumatoid arthritis treatment. a** Coordination geometry around the $Mn^{III}$ center of human mitochondrial Mn-SOD (PDB: 1N0J). For better clarification the H atoms are omitted. **b** Coordination geometry around the $Fe^{III}$ center in the heme group of human erythrocyte catalase (PDB: 1DGF). H atoms are also omitted. **c** Chemical structure of the 2D MOF ZMTP nanosheet, which fixes and periodically assembles a metalloporphyrin $MnTCPP^+$ in a regular, ordered, and symmetric pattern. **d** Building units of ZMTP nanosheets, $Zn(COO)_4$ paddlewheel metal nodes and $MnTCPP^+$ ligand. The latter has a $Mn^{III}$ center with a coordination environment being analogous to those of the active metal sites of Mn-SOD and catalase. **e** Layered structure of ZMTP nanosheet. **f** Therapeutic mechanism of ZMTP nanosheet for nanocatalytic rheumatoid arthritis treatment. The 2D MOF with periodically arranged active $Mn^{III}$ sites mimics the activities of both SOD and catalase, catalyzing the disproportionations of $O_2^{\bullet-}$ and $H_2O_2$. In pathological sites, the nanosheets can downregulate the intracellular oxidative stress of M1 macrophages, promoting the polarization shift of these cells toward their anti-inflammatory M2 phenotype. This action reduces the apoptosis of mBMSCs. Additionally, the $Zn^{2+}$ released from the degraded nanosheets favors the upregulation of ALP in mBMSCs, promoting the formation of CaP (hydroxyapatite) components, finally accelerating biomineralization.

antioxidases in pathological sites suffers from their over-large sizes leading to short circulating half-life, limited cell permeability, and antigenicity[5]. Chemists are now trying to follow the Nature example to design new chemical structures with their coordination environments of metals being analogous to those of natural antioxidases for various catalytic-therapeutic purposes.

Manganese porphyrins and their Schiff base complexes are characterized with high metal-ligand stability, the accessibility of their metal centers toward multiple oxidation states, and their reactivity toward different types of ROS, which have provoked extensive in vivo studies[6]. Importantly, their Mn[III] centers are fixed in the N4-macroheterocycles and axially ligated with two water molecules, such coordination environments are geometrically analogous to those of the active metal sites of Mn-SOD and catalase, making this category of molecules feature both SOD- and catalase-like activities[6,7]. Such a catalytic functionality of manganese porphyrins are highly dependent on the interactions between the Mn center and porphyrin skeleton, i.e., porphyrin-to-Mn $\sigma$ donation and Mn-porphyrin $\pi$ mixing[8]. Several manganese porphyrins, such as Mn[III] meso-tetrakis(4-carboxyphenyl)porphyrin (MnTCPP[+]), have been applied for treating of oxidative stress injuries and inflammation[9,10]. However, the catalytic activity of MnTCPP[+] is quite low due to the negative metal-centered redox potential ($E_{1/2} = -194$ mV versus normal hydrogen electrode (NHE))[11,12], as its four benzoyloxys are strong electron-donating groups after deprotonation.

Metal–organic framework (MOF) is an emerging type of crystalline porous material, which is prepared based on the chemical mechanism that metal ions or clusters can be coordinated by the carboxy groups of polytopic organic ligands[13]. Therefore, it is expected that the benzoyloxy-containing MnTCPP[+] can also act as an organic ligand to coordinate extrinsic metal ions and form a MOF structure, where the active Mn[III] sites of MnTCPP[+] are arranged in a highly regular, ordered, and symmetric pattern. Importantly, the positively-charged metal ions can counteract the negative charge of benzoyloxys, therefore elevating the metal-centered redox potential of MnTCPP[+] and its consequent catalytic activity. It is noted that for catalytic-therapeutic applications, the active metal sites should be exposed on the outer surface of materials as much as possible for maximized catalytic activity.

Two-dimensional nanomaterials have been extensively applied in biomedical fields recently, owing to their unique planar topography that results in various excellent physiochemical properties[14]. This category of nanomaterials is highly suitable and competent for nanocatalytic therapeutic purposes as their ultrathin thickness would maximize the exposure of active metal sites on the surface[15]. Therefore, it expected that the 2D MnTCPP[+]-based MOF nanosheets can be prepared through a rational chemical approach for achieving highly efficient nanocatalytic antioxidative therapy.

In this work, we prepare 2D manganese porphyrin-based MOF nanosheets and use them as artificial antioxidases (antioxidant nanozymes) for nanocatalytic rheumatoid arthritis treatment. The 2D MOF is synthesized through a coordination reaction between the benzoyloxys of MnTCPP[+] and Zn[2+], forming an ordered and symmetric structure periodically assembling active Mn[III] sites (Fig. 1c). The secondary building units (SBUs), Zn2(COO)4 paddlewheel metal nodes, are catalytically non-active, but play a role of fixing the catalytically active MnTCPP[+] ligands (Fig. 1d) in the layered structure stacked in an AB packing pattern (Fig. 1e). During the synthesis, the addition of surfactant polyvinylpyrrolidone (PVP) leads to the anisotropic growth of MOF favoring the formation of an ultrathin 2D structure.

The Zn-MnTCPP-PVP (denoted ZMTP) nanosheets mimic the activities of both SOD and catalase, to catalyze the disproportionations of both $O_2^{\bullet-}$ and $H_2O_2$, respectively. This action of ZMTP nanosheet at the pathological site of rheumatoid arthritis mitigates local oxidative stress, promoting the transition of proinflammatory M1 macrophages toward anti-inflammatory M2 phenotype, therefore depressing the inflammation-related apoptosis of bone mesenchymal stem cells (BMSCs) (Fig. 1f). More importantly, as Zn is also the catalytic center of alkaline phosphatase (ALP), a key phosphomonoesterase during osteogenic differentiation, the $Zn^{2+}$ released from the degrading ZMTP nanosheets can promote ALP expression, thereby facilitating biomineralization. The adjuvant-induced arthritis (AIA) animal model further indicates the high antiarthritic efficacy of ZMTP nanosheets, demonstrating the high feasibility of this 2D MOF for nanocatalytic rheumatoid arthritis treatment.

## Results

**Synthesis and characterization.** Ultrathin 2D ZMTP MOF was prepared by a surfactant-assisted bottom-up solvothermal approach[16–18], which features a high-yield and a large-scale production of this 2D material. The monochloride of MnTCPP[+] and Zn(NO3)2 were dispersed and reacted in a mixture of two hot polar solvents, N,N-dimethylformamide (DMF) and ethanol, while pyrazine was also added as a weak base to promote the deprotonation of benzoyloxys of MnTCPP[+] favoring its coordination with $Zn^{2+}$ (Fig. 2a). Additionally, PVP, an amphiphilic polymer extensively used as a surfactant, was also added in the mixture not only for stabilizing the material in polar solvents but also for acting as a capping agent to enable the anisotropic growth of the material into an ultrathin 2D structure. The carbonyl groups (C=O) of pyrrolidone rings of PVP can weakly and axially coordinate with the Zn atoms in the SBUs, Zn2(COO)4 paddlewheel clusters[19], while the trans-coordination of PVP with the two Zn atoms in each SBU creates a sandwiched structure with a thin interlayer for directing the growth of this 2D material. For biomedical application, the PVP on the surface of the 2D structure also elevates the dispersity and stability of the material in saline solution.

In the ultrathin 2D MOF, the organic linker MnTCPP[+] provides a fully exposed catalytic Mn[III] center, while the two Zn atoms in each paddlewheel metal node connect four carboxy groups though bridge-type coordination, enabling periodic assembly of MnTCPP[+] in a 2D structure with a space group of I4/mmm (isoreticular to the reported PPF-1[20]). Through the coordination, the negative charge of the peripheral four benzoyloxy groups of manganese porphyrin is counteracted, thus the redox potential of Mn[III] center is elevated, benefitting ROS-scavenging reactions. The Zn atoms in SBUs also provide anchoring sites for absorbing PVP, favoring the formation of highly dispersed 2D structure. In the following section, we will further discuss the role of Zn specie in the 2D MOF in promoting the expression of ALP during osteogenic differentiation. Therefore, in this study all the components of the 2D MOF (MnTCPP[+], Zn and PVP) play their respective roles in benefitting subsequent catalytic-therapeutic applications, in a versatile and synergistic manner.

The synthesized 2D MOF has a lateral size of over 3 μm according to the transmission electron microscopy (TEM) image (Supplementary Fig. 2). To tailor the 2D sheets within a nanodimension, probe sonication was applied to reduce their lateral sizes through an ultrasonic mechanical force (Fig. 2a). TEM image shows that the as-prepared nanosheets after probe sonication are highly dispersed with a lateral size of <300 nm (Fig. 2b), with a smooth surface and a rough margin (Fig. 2c), which is further evidenced by scanning electron microscopy (SEM) imaging (Fig. 2d). Atomic force microscopy (AFM) image and corresponding cross-sectional analysis manifest the <4-nm

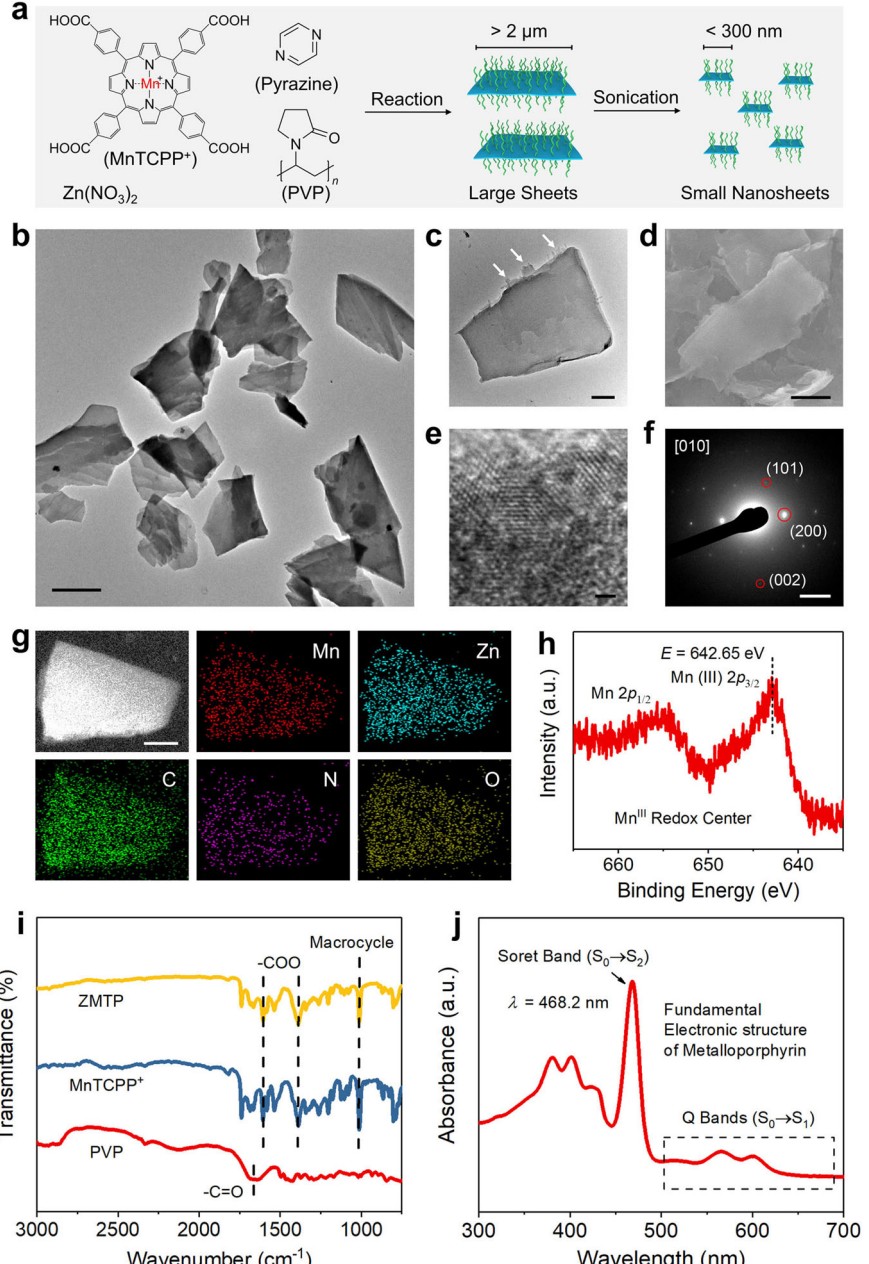

**Fig. 2 Synthesis and characterizations of ZMTP nanosheets. a** Synthesis of the 2D MOF nanosheets through a solvothermal reaction to produce micrometre-sized large sheets, followed by probe sonication to prepare small nanosheets of <300 nm in lateral size. **b** TEM image of highly dispersed ZMTP nanosheets. Scale bar, 100 nm. **c** TEM image of a representative ZMTP nanosheet. White arrows indicate the rough edge of the 2D MOF. Scale bar, 20 nm. **d** SEM image of a representative ZMTP nanosheet showing its smooth surface. Scale bar, 50 nm. **e** High-resolution TEM image of a ZMTP nanosheet. Scale bar, 1 nm. **f** SAED pattern of a ZMTP nanosheet collected along the [010] axis. Scale bar, 1 nm⁻¹. **g** High-angle annular dark-field (HAADF) image as well as element mappings (Mn, Zn, C, N, O) of a representative ZMTP nanosheet. Scale bar, 50 nm. A representative image of three replicates is shown. **h** Mn 2p XPS spectrum of the ZMTP nanosheet sample. **i** FTIR spectra of PVP, MnTCPP⁺ and ZMTP. **j** UV-vis absorption spectrum of the aqueous solution of ZMTP nanosheets. Source data are provided as a Source Data file.

thickness of ZMTP nanosheets (Supplementary Fig. 3). As the theoretical interlayer distance of ZMTP is 0.93 nm according to the reported isostructural PPF-1 model[20], it can be inferred that the layer number of ZMTP nanosheet is around 4.

Different from most MOFs, which are unstable under the irradiation of high-energy electron beam during the observation by high-resolution TEM, distinct lattice fringes can be observed in the high-resolution TEM image of the ZMTP nanosheet (Fig. 2e), revealing its high and robust crystallinity, in agreement with the result of X-ray diffraction (XRD) pattern (Supplementary Fig. 4).

Selected-area electron diffraction (SAED) pattern of the nanosheet collected along the [010] axis shows the diffraction spots attributed to the (101), (200) and (002) planes (Fig. 2f), indicating its tetragonal structure, which is identical to that of this category of pristine MOF[17], suggesting that the structure of the material has been maintained during probe sonication. Element mappings and energy-dispersive spectroscopy (EDS) profiles demonstrate the presence and uniform distribution of Mn, Zn, C, N, and O elements in the nanosheet (Fig. 2g and Supplementary Fig. 5), while the result of X-ray photoelectron spectroscopy

(XPS) further indicates the +3 oxidation state of Mn center in the nanosheet (Fig. 2h), in consistence with those of the Mn centers in free manganese porphyrin molecules in their resting state[11], therefore probably presenting similar redox properties.

Fourier transform infrared (FTIR) spectrum of ZMTP shows two absorption bands at 1605.9 cm$^{-1}$ and 1389.5 cm$^{-1}$ assigned to the vibration of carboxylate group ($-$COO) of MnTCPP$^+$ (Fig. 2i), as well as an absorption band at 1013.9 cm$^{-1}$ attributed to the vibration of macroheterocycle of MnTCPP$^+$. The absorption band at 1661.4 cm$^{-1}$ attributed to the vibration of carbonyl group of pristine PVP is not distinct in the FTIR spectrum of ZMTP, which may result from the weak coordination between the carbonyl group of PVP and the Zn atoms in SBUs that leads to the red shift of the absorption band[17]. The presence of PVP on the nanosheet significantly improves the dispersibility and stability of the 2D MOF in deionized water and phosphate buffer saline (PBS), and no precipitate can be observed in 12 h of dispersion (Supplementary Fig. 6). Continuously washing the material to remove PVP will lead to the aggregation of the nanosheets in PBS, further manifesting the important role of PVP in maintaining the stability of nanosheets in solution. UV-vis absorption spectrum of ZMTP in aqueous solution shows a characteristic Soret band ($\lambda = 468.2$ nm) and two Q bands (500–700 nm) of metalloporphyrin (Fig. 2j), indicating that the fundamental metalloporphyrin structure of MnTCPP$^+$, i.e., Mn-N$_4$-macroheterocycle, has been well-preserved in the nanosheets. The two types of characteristic bands result from two $\pi$-$\pi^*$ transitions of porphyrin ring ($a_{1u} \rightarrow e_g^*$, $a_{2u} \rightarrow e_g^*$) after a configuration interaction, the transition dipoles add ($S_0 \rightarrow S_2$) leads to the emergence of Soret band while the transition dipoles cancel ($S_0 \rightarrow S_1$) enables the formation of two Q bands after molecular orbital degeneration[21].

**Catalytic mechanism and efficiency.** As the coordination environment of Mn$^{III}$ center in MnTCPP$^+$ is analogous to those of functional metal centers of Mn-SOD and catalase, following the structure-activity relationship, it is inferred that the unique structure mimicry of MnTCPP$^+$ ligand will endow this 2D MOF with SOD- and catalase-like activities. The detailed catalytic process at the active metal sites of natural Mn-SOD and catalase will be elucidated, for a better exploration of the catalytic mechanism of ZMTP nanosheets, such as valence transition of active Mn sites and the accompanied electron and proton transfers.

The disproportionation of O$_2$$^{\bullet-}$ at the Mn$^{III}$ site of Mn-SOD is based on a "ping-pong" mechanism involving Mn$^{III}$/Mn$^{II}$ conversion and hydroxo/aqua ligand shifting[22]. One O$_2$$^{\bullet-}$ first coordinates the Mn$^{III}$ center in a trans position to Asp 160 and thereafter reduces Mn$^{III}$ to Mn$^{II}$, after which one O$_2$ molecule is generated (Supplementary Table 1). During the half reaction, the axial ligand, OH$^-$ accepts one H$^+$ to form a H$_2$O molecule for maintaining electric neutrality. The next O$_2$$^{\bullet-}$ anion then coordinates Mn$^{II}$ center and accepts the proton from the axial H$_2$O ligand, forming an unstable Mn$^{II}$-OOH$^-$ complex that will further accept one proton from ambient environment and thereafter self-dissociate into Mn$^{III}$ and H$_2$O$_2$. Similarly, during the disproportionation of H$_2$O$_2$ at the Fe$^{III}$ site of catalase, two H$_2$O$_2$ molecules are also required for completing the catalytic process, one of which first coordinates the metal center in the trans position to Tyr 354 to enable two-electron oxidation of the Fe$^{III}$ site, leading to the formation of an oxoferryl porphyrin cation radical (P$^{\bullet+}$-Fe$^{IV}$ = O, P represents PpIX) and one H$_2$O molecule (Supplementary Table 1). The other H$_2$O$_2$ molecule then favors the protonation of oxoferryl group and subsequent two-electron reduction, making the iron porphyrin complex

return to its resting state (P-Fe$^{III}$), after which one H$_2$O and one O$_2$ molecules are released. The distal His favors proton transfer during the whole reaction process (Supplementary Fig. 7), while Tyr 354 promotes the stabilization of oxidation state of Fe center at +4 (rather than +5) by enabling the generation of porphyrin cation radical, therefore reducing the energy required for reaction.

MOF biomimicry is an emerging and attractive research field, which constructs active open metal sites in MOF with coordination geometries analogous to those of the functional metal sites of natural enzymes, thus achieving structural and functional mimicking to these enzymes using MOF[23–25]. Most of the reports in this field are based on the modification of the coordination environments of metals in SBUs, in this work we instead directly take advantage of the unique structural characteristic of the organic linker MnTCPP$^+$, i.e., the dual antioxidase-analogous coordination environment of Mn$^{III}$ center, for constructing a MOF nanosheet with both SOD- and catalase-like activities. The ultrathin 2D structure further offers much more accessible active sites than those of nanocatalysts with other morphologies. The ZMTP nanosheet can be considered as an ordered self-assembly of numbers of MnTCPP$^+$ (as catalytic units) interconnected by SBUs, and the Zn atoms in the SBUs regulate the redox potential of Mn$^{III}$ centers by counteracting the negative charge of peripheral four benzoyloxy groups of MnTCPP$^+$ (the metal-centered redox potential of natural Mn-SOD is around 300 mV versus NHE[11], Supplementary Fig. 8). Therefore, based on the reported redox characteristics of manganese porphyrin molecules, we probe the reaction mechanism of ROS scavenging in each catalytic unit of ZMTP nanosheet.

In aqueous solution the Mn$^{III}$ center of manganese porphyrin in resting state is trans-axially coordinated by two solvent molecules (H$_2$O or/and OH$^-$)[26]. According to the electric neutrality of the nanomaterial, it can be inferred that one H$_2$O and one OH$^-$ are axially ligated on the Mn$^{III}$ center to counteract its one unit of positive charge, forming a hydroxo-aquaMn(III) porphyrin (Fig. 3a). Such a resting state of ZMTP nanosheets can be depicted as $\{Zn_2[(H_2O)(OH)Mn^{III}TCPP]\}_{(x,y,z)}$ ($x$, $y$, $z$ represent the numbers of periodic units in the horizontal, longitudinal and axial directions of nanosheets, respectively, $\iiint dxdydz = n$) based on the stoichiometric proportion of each component (here the surfactant PVP is not included). The disproportionation of O$_2$$^{\bullet-}$ by manganese porphyrin is based on the conversion between Mn$^{III}$ and Mn$^{II}$ accompanied by outer-sphere proton-coupled one-electron transfer[27], and the redox potential of metal center is a dominating factor for enabling the reaction in accordance with the Marcus equation[28]. During the catalysis a transition state of ZMTP nanosheet $\{Zn_2[(H_2O)Mn^{II}TCPP]\}_{(x,y,z)}$ forms, where the mono-aquaMn(II) is pentacoordinated forming a tetragonal structure[27]. Comparatively, during the disproportionation of H$_2$O$_2$ by the nanosheet, which is based on the conversion between Mn$^{III}$ and Mn$^V$ accompanied by inner-sphere proton-coupled two-electron transfer[27], a new transition state of ZMTP nanosheet $\{Zn_2[(O)(OH)Mn^VTCPP]\}_{(x,y,z)}$ with oxo-hydroxoMn(V) centers will form. At the beginning of reaction, one H$_2$O$_2$ molecule first deprotonates to form a hydroperoxide anion (HOO$^-$), which then replaces one axial hydroxo ligand of hydroxo-aquaMn(III) center to form an hydroperoxo-aqua complex (Supplementary Fig. 9)[29]. Once coordinated, both the redox potentials of Mn center and HOO$^-$ specie will be changed, thus the metal-centered redox potential of MnTCPP$^+$ in the nanosheet is not a dominant factor for H$_2$O$_2$ decomposition. The detailed reactants, intermediates, and products of the SOD- and catalase-mimicking catalytic reactions by the 2D MOF are presented in Fig. 3a.

Electrochemical measurements have been used for further evidencing the redox properties of the ZMTP nanosheets. Cyclic voltammetry (CV) results show that, compared with single Mn$^{2+}$, both the positions of anodic and cathodic peak potentials

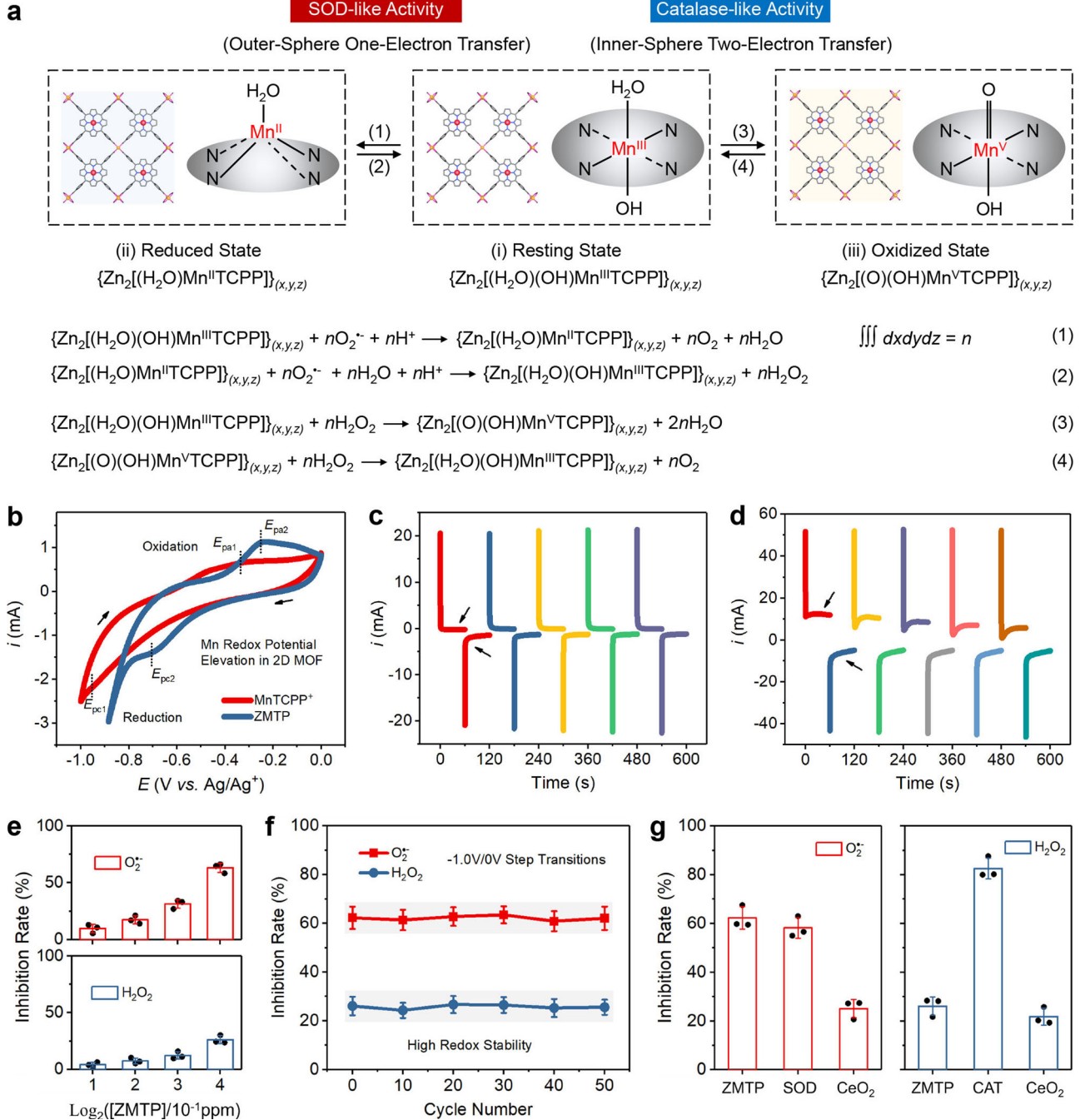

**Fig. 3 Catalytic mechanism of ZMTP nanosheet and its antioxidative efficiency. a** Schematic for the interconversion among resting, reduced, and oxidized states of ZMTP nanosheet, as well as the detailed chemical reactions in each step of conversion. For each state of ZMTP nanosheet, the stereoscopic structure of MnTCPP$^+$ is also simplified as a Mn-N$_4$ structure as well as axial ligand(s) for clarification. The detailed chemical formula of ZMTP nanosheet in each state has been presented as a triple integral of corresponding periodic unit over the whole region of nanosheet. **b** CV curves of MnTCPP$^+$ ([Mn] = 1 mM) and ZMTP ([Mn] = 1 mM) in electrolyte solutions investigating the redox responses. $E_{pa1}$ and $E_{pa2}$ correspond to the anodic peak potentials of MnTCPP$^+$ and ZMTP in electrolyte solutions, respectively, while $E_{pc1}$ and $E_{pc2}$ correspond to the cathodic peak potentials of MnTCPP$^+$ and ZMTP in electrolyte solutions, respectively. **c** CA curves showing the limit currents of ZMTP in electrolyte solution during the five successive periods of transition of two potentials (high potential: −0.5 V; low potential: −1.0 V). **d** CA curves showing a changed limit current characteristic of ZMTP in electrolyte solution during the five successive periods of transition of two newly-established potentials (high potential: 0 V; low potential: −1.0 V). **e** Inhibition rates of O$_2^{•-}$ and H$_2$O$_2$ by varied concentrations of ZMTP nanosheets for 4 h, corresponding to the SOD- and catalase-like activities. Data are expressed as mean ± SD (N = 3 independent experiments). **f** Evaluation on the redox stability of ZMTP nanosheets. The ZMTP (1.5 ppm)-containing electrolyte solution was subjected to the consecutive actions and transitions of two step potentials (high potential: 0 V; low potential: −1.0 V; 50 cycles). An aliquot was obtained from the solution after each ten cycles for measuring the SOD- and catalase-like activities of ZMTP nanosheets. Data are expressed as mean ± SD (N = 3 independent experiments). **g** Comparison of the catalytic activity of ZMTP nanosheets with those of nature antioxidases (SOD and catalase) and nanozyme (CeO$_2$ nanoparticle). The concentrations of all these agents are 1.5 ppm. CAT, catalase. Data are expressed as mean ± SD (N = 3 independent experiments). Source data are provided as a Source Data file.

($E_{pa}$ and $E_{pc}$) of ZMTP have shifted to the middle region across the voltage window (Supplementary Fig. 10), indicating that the Mn center in the 2D MOF nanosheet has been activated and both oxidation and reduction reactions are favored. Additionally, compared with single MnTCPP$^+$, both the positions of $E_{pa}$ and $E_{pc}$ of ZMTP have shifted positively (Fig. 3b), further demonstrating that the Zn component in the 2D MOF upregulates the redox potential of the Mn center of manganese porphyrin, benefiting the redox reaction processes. The electrolyte solution used in this study is mildly acidic with a pH value of 6.6, exactly the mean pH value of rheumatoid joints[30]. This experimental design favors further prediction of redox properties of the 2D MOF in rheumatoid joints region. However, in this electrochemical technique of CV, the Mn$^V$ species cannot be accessed directly[31], as the voltage during measurement is always changing. The redox reaction here only involves the conversion between hydroxo-aquaMn(III) and mono-aquaMn(II) in the 2D MOF (Supplementary Table 2).

Chronoamperometry (CA) applies a continuous potential over a period for recoding the limit current output, while allows the alternative transitions between two step potentials for promoting redox cycling, which is named as double potential step chron-oamperometry (DPSCA). This electrochemical method here is used for further comprehensive evaluation of the redox properties of ZMTP nanosheets, especially valence changes of Mn centers, by monitoring the limited current. The DPSCA measurement of ZMTP in electrolyte solution was conducted under the application of five consecutive cycles of transitions between a higher potential of $-0.5$ V and a lower potential of $-1.0$ V with a period of 120 s for each cycle (Supplementary Fig. 11a), which were set based on the $E_{pa}$ and $E_{pc}$ of ZMTP measured by CV for only enabling the reduction reaction of the 2D material. The limit current responses are almost identical during the five periods (Fig. 3c), which may result from the fast reduction of hydroxo-aquaMn(III) center of ZMTP to monoaquaMn(II) in the first period forming $\{Zn_2[(H_2O)Mn^{II}TCPP]\}_{(x,y,z)}$ (Supplementary Table 3), thus presenting the limit current responses of the reduced 2D MOF during the five periods. This process corresponds to the half reaction of equation (1) in Fig. 3a. Alternatively and interestingly, when fresh ZMTP-containing electrolyte solution was subjected to the renewed two potentials for five consecutive cycles (higher potential: 0 V, lower potential: $-1.0$ V, period: 120 s) (Supplementary Fig. 11b), a minor increase of positive current can be observed in each period after initial current collapse (Fig. 3d). This may result from the generation of oxo-hydroxoMn(V) in the 2D MOF during the continuous action of the higher potential of 0 V (Supplementary Table 4), which leads to the additional limit current response. This chemical process can be assigned to the equation (3) in Fig. 3a. Accordingly, the response of negative limit current in each period is related to the reduction of the oxo-hydroxoMn(V) centers of 2D MOF to hydroxo-aquaMn(III) and to monoaquaMn(II), corre-sponding to the half reactions of (4) and (1) in Fig. 3a, while the positive limit current response in a next period is related to the oxidation of monoaquaMn(II) to hydroxo-aquaMn(III) and to oxo-hydroxoMn(V) forming $\{Zn_2[(O)(OH)Mn^VTCPP]\}_{(x,y,z)}$, relating to equation (2) and (3) in Fig. 3a. This round of CA measurement forms a closed redox cycle featuring quasi-reversible electron transfers, and negligible ZMTP nanosheets are degraded during the redox cycling (Supplementary Fig. 12 and Supplementary Table 5).

The generations of Mn$^{II}$ and Mn$^V$ species of 2D MOF were further evidenced by UV-vis spectra (Supplementary Fig. 13), which show a blue or red shift of the spectrum of ZMTP nanosheet after its reduction or oxidation during the above CA treatment[32,33]. It is noted that the redox reactions of 2D MOF involves the change of axial ligands and proton transfer, however, such a mechanism is hard to be analyzed by crystallography as the scattering ability of hydrogen atoms is extremely poor.

The catalytic activities of ZMTP nanosheets toward O$_2^{\bullet-}$ and H$_2$O$_2$ disproportionations have been further investigated, and dose-dependent increases of the inhibition rates of both O$_2^{\bullet-}$ and H$_2$O$_2$ have been recorded (Fig. 3e). The catalytic activity of ZMTP nanosheet toward H$_2$O$_2$ disproportionation is lower than that toward O$_2^{\bullet-}$ disproportionation, which may be due to the harder inter-conversion between Mn$^{III}$ and high-valence Mn$^V$ than Mn$^{III}$/Mn$^{II}$ cycling. Single MnTCPP$^+$ only presents insignif-icant SOD-like activity (Supplementary Fig. 14a), as its redox potential is relatively low for initiating the first step of O$_2^{\bullet-}$ disproportionation[11]. This manganese porphyrin fails to present catalase-like activity (Supplementary Fig. 14b), which results from its 3 units of negative charges after deprotonation that blocks the axial coordination of HOO$^-$ anion. Additionally, Zn$^{2+}$ was confirmed to be non-catalytic. However, this divalent cation upregulates the redox potential of Mn center in the 2D MOF by counteracting the negative charges of benzoyloxys of MnTCPP$^+$, therefore significantly elevating the catalytic activities.

For further confirming the functions of different components of ZMTP nanosheet, two isostructural 2D MOFs, Zn-ZnTCPP-PVP and Zn-FeTCPP-PVP nanosheets, which are only different from ZMTP in the type of metal center of porphyrin, were also prepared through the same procedure for comparison with ZMTP. Zn-ZnTCPP-PVP possesses a redox-inert Zn center in the porphyrin, while Zn-FeTCPP-PVP has a Fe center that may show redox characteristic. The antioxidative activities of the two isostructural 2D MOFs have also been evaluated (Supplementary Fig. 15). No catalytic antioxidative activity has been detected in solution containing Zn-ZnTCPP-PVP, indirectly suggesting that components other than Mn center in ZMTP nanosheet are non-catalytic. Additionally, the Zn-FeTCPP-PVP nanosheet also failed to present detectable SOD- or catalase-like activities, indicating that the type of transition metal center of porphyrin is a prerequisite for catalyzing a series of desired redox reactions that mimic the catalytic activities of both SOD and catalase.

To further investigate the redox stability of the 2D material during catalysis, the ZMTP nanosheets were also dispersed in electrolyte solution and subjected to the step potential transitions between 0 V and $-1.0$ V with a period of 120 s in DPSCA measurements (Supplementary Fig. 16). An aliquot was obtained from the solution after each ten cycles of step potential transitions for measuring the SOD- and catalase-like activities, and the 2D MOF shows a high redox stability even after 50 cycles of potential transitions. Insignificant degradation of ZMTP nanosheet has also been observed during the process (Supplementary Fig. 17 and Supplementary Table 5).

The catalytic performance of ZMTP nanosheet was further compared with natural antioxidases (SOD and catalase) and CeO$_2$ nanoparticles as inorganic nanozymes possessing both SOD- and catalase-like activities[34–36]. The O$_2^{\bullet-}$-inhibition capability of ZMTP is stronger than that of natural SOD to a certain extent (Fig. 3g), but the H$_2$O$_2$-inhibition efficiency of ZMTP is much lower than that of natural catalase, as in catalase the oxoferryl porphyrin cation radical (P$^{\bullet+}$-Fe$^{IV}$=O) (rather than Fe$^V$) can form under the assistance of Tyr 354 residue, which favors the valence transition of Fe center during H$_2$O$_2$ decomposition. Moreover, the distal histidine residue of catalase can also promote proton transfer during the redox reaction. Nevertheless, the catalytic activity of ZMTP nanosheet is significant and substan-tially higher than the typical nanozyme CeO$_2$ nanoparticle, of which the conversion of Ce$^{4+}$ to Ce$^{3+}$ is a rate-limiting step[37]. The antioxidative activity of ZMTP was further compared with those of ascorbic acid, gallic acid, and reduced glutathione, and their concentrations required for inhibiting half amounts of ROS (O$_2^{\bullet-}$ and H$_2$O$_2$) were determined (Supplementary Table 6). ZMTP nanosheet plays a role of nanocatalyst while these

antioxidants act as reactants in the redox reaction, thus the required dose of ZMTP is much lower than those of ascorbic acid, gallic acid, and reduced glutathione, which can only scavenge ROS stoichiometrically.

**Long-term degradability**. The above electrochemical measurements also reveal that the 2D MOF is stable during short-term redox reactions. The long-term degradability of ZMTP nanosheets were further investigated by dispersing them in mild acidic PBS (pH = 6.6). The effect of fluid stress in a solution is one of the main factors determining the degradation of nanomaterial. According to the well-known Bernoulli's equation in physics, if the flow velocities of solution over the two planar surfaces of nanosheet are different from each other, the fluid stress perpendicular to the two sides of the nanosheets will be also different, which will lead to the axial weak deformation of the ultrathin nanosheets, by which pores may be created on the nanosheets (Fig. 4a). These pores as structural defects will further promote the degradation of nanosheets.

TEM images reveal that no significant morphological change of nanosheets can be observed after dispersing in mild acidic PBS for 6 h (Fig. 4b, c), demonstrating the short-term stability of the 2D MOF in solution. However, after dispersing in mild acidic PBS for 12 h, very small pores were generated on the nanosheet (Supplementary Fig. 18), which enlarges in 24 h of degradation (Fig. 4d and Supplementary Fig. 19). Both SAED pattern and high-resolution TEM image show the significantly weakened crystallinity of the nanosheet in 24 h of degradation (Fig. 4d and Supplementary Fig. 20), which may contribute to accelerated degradation of the nanosheet. Only small fragments of the nanosheets can be observed in 48 h of degradation (Fig. 4e), while negligible debris were left after dispersed in mild acidic PBS for 96 h (Fig. 4f). UV-vis spectra of ZMTP nanosheets also demonstrate the minor degradation of the 2D material during the initial 6 h of dispersion in mild acidic PBS (Fig. 4g), while in the subsequent time duration the degradation of ZMTP accelerates. The time-course release of Zn from the nanosheets has also been detected (Fig. 4h), confirming the two-stage

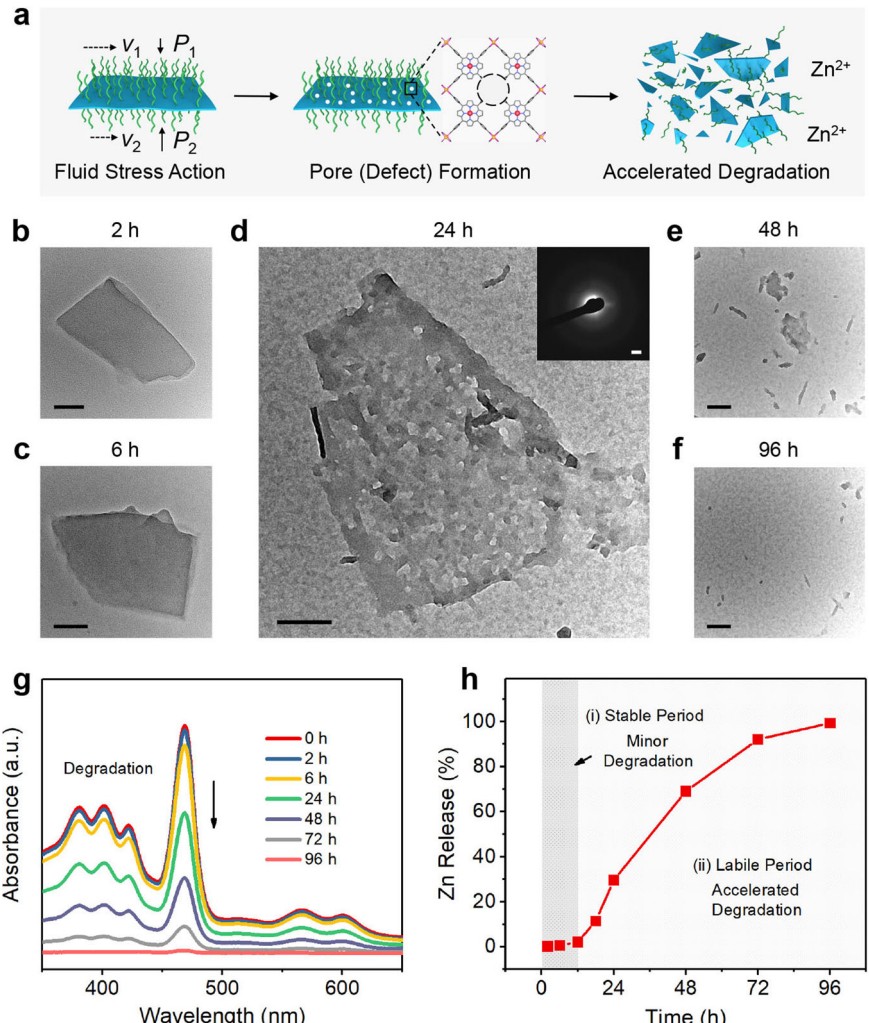

**Fig. 4 Long-term degradability of ZMTP nanosheets in response to fluid stress. a** Scheme for the degradation process of ZMTP nanosheets during a long time period. The different fluid flow velocities on the two planar surfaces of a nanosheet in solution create differential fluid stress perpendicular to the two sides of the nanosheet, resulting in the axial weak deformation of the thin nanosheet and subsequent formation of pores. These structural defects will further promote the degradation of the whole nanosheets and the release of $Zn^{2+}$. **b–f** TEM images of ZMTP nanosheets after degradation in mild acidic PBS for varied time durations. Scale bars, 30 nm. Inset in (**d**) is the SAED pattern of the degrading nanosheet. Scale bar, 2 $nm^{-1}$. A representative image of three replicates is shown. **g** UV-vis absorption spectra of mild acidic PBS containing ZMTP nanosheets after degradation for varied time durations. **h** Time-dependent Zn release from the degrading ZMTP nanosheets. During the first 12 h only a minor amount of Zn was released, followed by a significantly accelerated release during the later period. Source data are provided as a Source Data file.

degradation behavior of the nanosheet: an initial stable period of minor degradation of ZMTP, and subsequent labile period of significantly accelerated degradation of the 2D MOF. From the viewpoint of subsequent biological application of the nanosheets, such a degradation characteristic favourably guarantees the lasting catalytic activity of ZMTP during the antioxidative therapy, as well as the long-term biosafety of the material by subsequent degradation and clearance.

**Cellular anti-inflammatory efficiency.** The progression of rheumatoid arthritis is related with various biochemical cues especially oxidative stress[4]. The overexpression of ROS leads to the polarization of macrophages toward pro-inflammatory M1 phenotype, which secrete interleukin-6 (IL-6), interleukin-1β (IL-1β), tumor necrosis factor-α (TNF-α) and other inflammation cytokines possessing negative effects on the physiological activities of BMSCs[38]. It is conceived that the as-constructed ZMTP nanosheet can catalytically scavenge intracellular ROS of M1 macrophages to favor their polarization toward anti-inflammatory M2 macrophages, finally benefiting the treatment of rheumatoid arthritis.

To evaluate the cellular anti-inflammatory performance of ZMTP nanosheets, a murine mononuclear macrophage cell line Raw 264.7 was used for investigation in this work, and lipopolysaccharide (LPS, from Gram-negative bacteria[39]) was used to promote the polarization of pristine Raw 264.7 cells toward pro-inflammatory M1 macrophages. The M1 macrophages were then incubated with culture medium containing ZMTP nanosheets, and confocal laser scanning microscopy (CLSM) images show that the nanomaterials could be internalized by M1 macrophages in 2 h of incubation (Fig. 5a). The 2D MOF shows negligible toxicity in M1 macrophages even in 48 h of co-incubation, as determined by cell counting kit-8 (CCK-8) assay (Fig. 5b). For revealing the cellular antioxidative performance of ZMTP, an ROS probe 2′, 7′-dichlorofluorescein diacetate (DCFH-DA) was applied for monitoring cellular ROS as well as semiquantitative evaluation. This probe can react with ROS to form 2′, 7′-dichlorofluorescein (DCF, green fluorescence)[40], and CLSM images show significant green signal in M1 macrophages (LPS group, Fig. 5c), compared with pristine Raw 264.7 cells (control group) where negligible signal could be monitored. ZMTP nanosheet shows a distinct ROS-scavenging effect in M1 macrophages, while MnTCPP[+] and Zn[2+] present negligible cellular antioxidative effect. Flow cytometry further reveals a dose-dependent antioxidative feature of ZMTP in M1 macrophages (Fig. 5d and Supplementary Fig. 21).

Significant morphological changes of M1 macrophages could be observed after treatment with ZMTP nanosheets (Fig. 5e), characteristic of tentacle shrinkage and the formation of a rough morphology, suggesting the polarization shift of the cells toward M2 phenotype (Fig. 5f). This phenomenon is associated to the intracellular catalytic ROS depletion by ZMTP nanosheets that promotes the phenotype transition (Fig. 5g). The expressions of mRNA of M1 macrophage markers, such as IL-6, IL-1β and TNF-α, which were determined by reverse transcription-polymerase chain reactions (RT-PCR) (Supplementary Table 7), were significantly downregulated after the treatment with 2D MOF (Fig. 5h), while the mRNA levels of M2 macrophage markers such as arginase-1 (Arg-1) and interleukin-10 (IL-10) were upregulated, further confirming the phenotype reprograming of the macrophages and the anti-inflammatory potential of ZMTP nanosheets. Comparatively, MnTCPP[+] or Zn[2+] alone show insignificant effects in regulating these cytokines based on the results of RT-PCR.

**Osteogenic differentiation promotion.** The regulation of inflammatory microenvironment and the level changes of

inflammation cytokines, may influence the downstream viability of BMSCs and their various physiological activities (especially osteogenic differentiation). In this study murine bone mesenchymal stem cells (mBMSCs) were used to incubate with M1 macrophages in a transwell system for investigating the effect of anti-inflammatory effects of ZMTP on the viability and osteogenic differentiation of mBMSCs (Fig. 6a). Nevertheless, the direct effect of ZMTP nanosheet on the physiological activities of mBMSCs were first investigated by incubating mBMSCs directly with ZMTP or single MnTCPP[+] or Zn[2+], and the CCK-8 results showed insignificant changes of the viability of mBMSCs even in 96 h of treatments in different groups (Fig. 6b), demonstrating that ZMTP alone has no significant effect on the viability of mBMSCs. Comparatively, after co-incubation of M1 macrophages and mBMSCs in the transwell system, a noticeable reduction of the viability of mBMSCs could be detected (Fig. 6c), while the additional ZMTP treatment can largely restore the activity of mBMSCs, attributing to the anti-inflammatory effect of the 2D material. Flow cytometry further revealed the apoptotic mechanism of mBMSCs in different groups (Fig. 6d and Supplementary Fig. 22). The co-incubation with M1 macrophages could lead to a considerable early apoptosis of mBMSCs (see Q3 quadrant), while ZMTP treatment has reduced the pro-apoptotic effect of M1 cells. MnTCPP[+] or Zn[2+] alone showed negligible effect on the reduction of mBMSC apoptosis, consistent with the experimental results of CCK-8 assay in Fig. 6c.

ALP is a key phosphomonoesterase during osteogenic differentiation especially biomineralization, which can catalyze the dephosphorylation of phosphate monoester substrates (DNA, RNA, proteins, and alkaloids) and the production of inorganic phosphate, providing abundant $PO_4^{3-}$ for the formation of mineralized nodules[41]. This metallophosphatase is a hybrid-type catalyst having two Zn[II] atoms as active metal sites with different coordination environments (Fig. 6e), while the formed phosphate between them is pentacoordinated and its equatorial plane bisects $Zn_I^{II}$-$Zn_2^{II}$ vector. The released $PO_4^{3-}$ will react with biological $Ca^{2+}$, leading to the formation of CaP (hydroxyapatite) and subsequent biomineralization (Fig. 6f). It is expected that $Zn^{2+}$ released from the 2D MOF during degradation can elevate the expression of ALP in mBMSCs. The mBMSCs were directly treated with MnTCPP[+], $Zn^{2+}$ and ZMTP for a prolonged time period (20 days), after which alizarin red S was used for stanning the mineralized nodules formed by mBMSCs (Fig. 6g)[42]. It could be observed that either $Zn^{2+}$ alone or ZMTP nanosheet can significantly promote the formation of mineralized nodules, demonstrating that the Zn species in the 2D MOF can indeed promote the osteoblast differentiation of mBMSCs and subsequent biomineralization. RT-PCR results also demonstrate that the level of ALP has been largely upregulated in mBMSCs co-incubated with M1 macrophages in the transwell system after treated with $Zn^{2+}$ and ZMTP (Fig. 6h), further evidencing that such a 2D MOF can release $Zn^{2+}$ to assist the expression of ALP in mBMSCs, finally contributing to biomineralization. The expressions of other types of mRNAs of osteogenic genes, such as osteopontin (OPN), osteocalcin (OCN), collagen type I (COL I) and Runt-related transcription factor 2 (RUNX 2), were also upregulated to varied extents in mBMSCs after ZMTP treatment. Due to the intrinsic characteristic of mBMSCs, no distinct morphological change could be found during osteogenic differentiation (Fig. 6i).

**In vivo antiarthritic efficacy.** Cellular experiments revealing the anti-inflammatory and pro-biomineralization performances of ZMTP nanosheet encouraged us to further explore its antiarthritic efficacy in vivo. Eight-week-old female Balb/c mice were

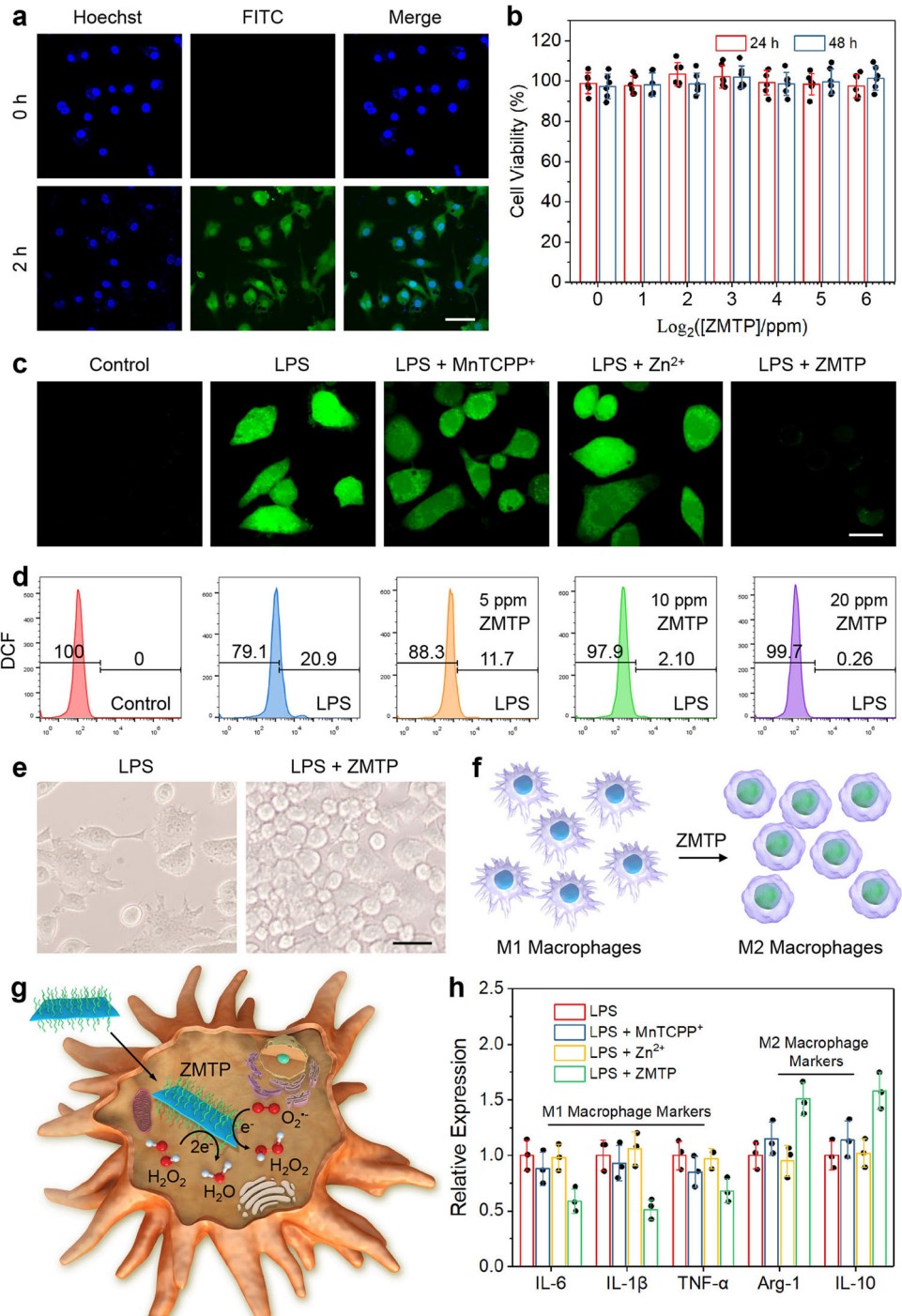

**Fig. 5 Antioxidative behavior and anti-inflammatory activity of ZMTP nanosheets in M1 macrophages. a** CLSM images indicating the internalization of FITC-labeled ZMTP nanosheets by M1 macrophages. A representative image of three replicates from each group is shown. Cells were also stained with bisbenzimide H 33258 fluorochrome (Hoechst) for nucleuses imaging. Scale bar, 200 μm. **b** Viability of M1 macrophages after treatment with varied concentrations of ZMTP for 24 h and 48 h. Data are expressed as means ± SD ($N = 6$ independent experiments). **c** CLSM images for imaging ROS in M1 macrophages after indicated treatments for 48 h, representative of 3 independent experiments. Primary Raw 264.7 cells without LPS activation were set as a control group. Scale bar, 100 μm. **d** Flow cytometric analysis evaluating the ROS depletion in M1 macrophages after treatment with varied concentrations of ZMTP nanosheets for 48 h, representative of 3 independent experiments. **e** Morphological changes of macrophages after treatment with ZMTP for 48 h, observed by optical microscope. A representative image of three replicates from each group is shown. Scale bar, 100 μm. **f** Schematic for the phenotype transition of M1 macrophages toward M2 ones. **g** Schematic for the cellular chemical mechanism of ZMTP nanosheet. **h** Expressions of the mRNAs of IL-6, IL-1β, TNF-α, Arg-1 and IL-10 in M1 macrophages after different treatments for 60 h, quantified by RT-PCR. The expressions of these markers in M1 macrophages without additional treatments were normalized as 1. Data are expressed as means ± SD ($N = 3$ independent experiments). Source data are provided as a Source Data file.

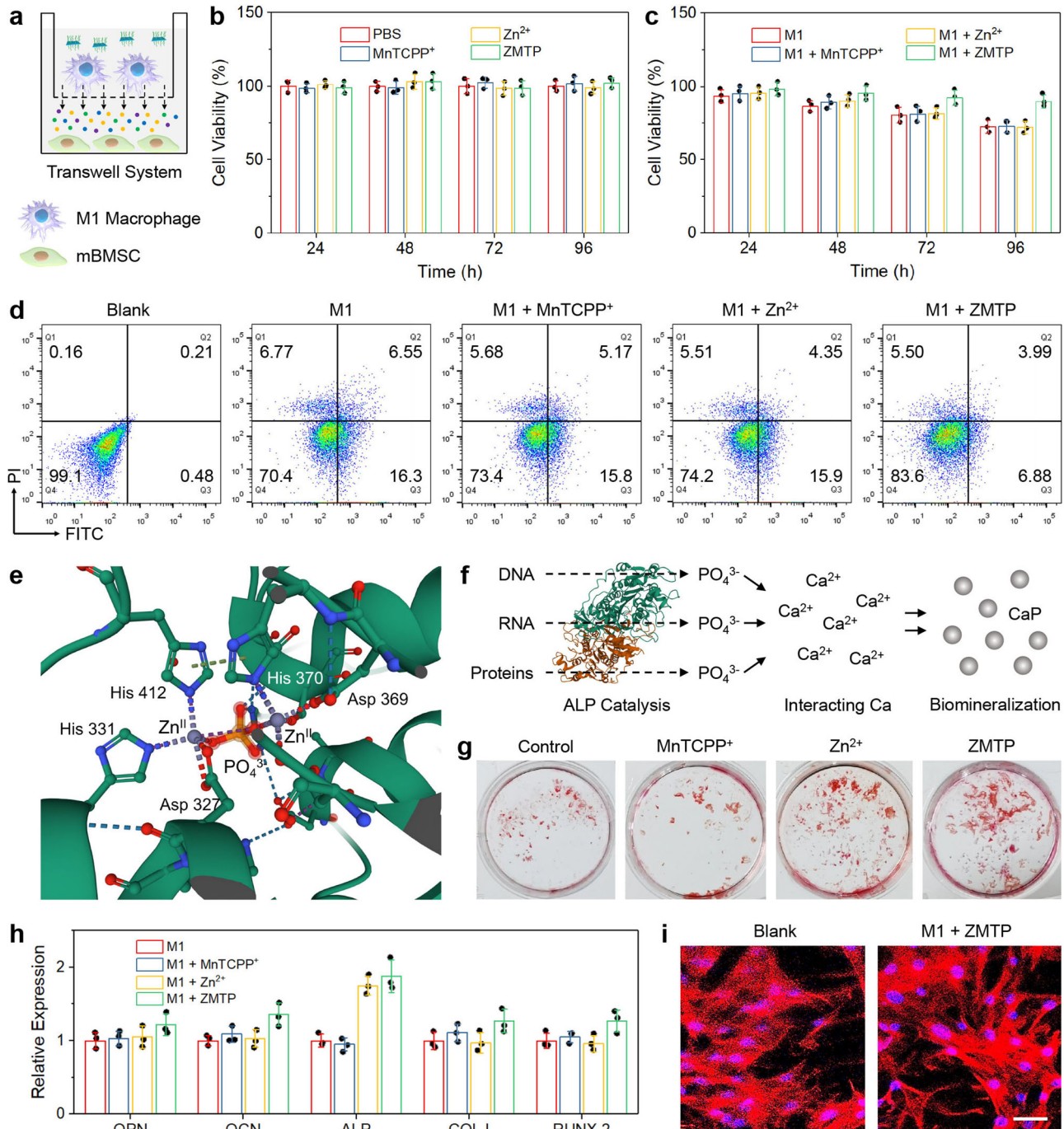

**Fig. 6 Osteogenic differentiation promoted by ZMTP nanosheets. a** Experimental setup for the transwell system. Chemicals can be exchanged through the filter membrane. **b** Viabilities of mBMSCs after direct treatments with different chemicals/materials for varied time durations without M1 macrophage co-incubation. Data are expressed as means ± SD (N = 3 independent experiments). **c** Viabilities of mBMSCs first co-incubated with M1 macrophages then treated with different chemicals/materials for varied time durations in the transwell system. Data are expressed as means ± SD (N = 3 independent experiments). **d** Flow cytometric analysis showing the change of apoptotic behaviors of mBMSCs co-incubated with M1 macrophages followed by treatments with different chemicals/materials for 48 h, representative of 3 independent experiments. Primary mBMSCs without M1 macrophage co-incubation and additional treatments were also used as a blank group. **e** Schematic of the coordination environment around the two $Zn^{II}$ active sites of ALP. One $PO_4^{3-}$ is produced between the two $Zn^{II}$ sites. H atoms are also omitted. **f** Chemical mechanism for ALP-promoted biomineralization. ALP catalyzes the dephosphorylation of phosphate monoester substrates (DNA, RNA, proteins, and alkaloids) and the production of inorganic phosphate, which will react with biological $Ca^{2+}$ to form CaP (hydroxyapatite), promoting biomineralization. **g** Alizarin red S staining of mineralized nodules of mBMSCs after directly treatment with MnTCPP+, $Zn^{2+}$ or ZMTP for 20 days. A representative image of three replicates from each group is shown. **h** Expressions of the mRNAs of OPN, OCN, ALP, COL 1, and RUNX 2 in mBMSCs after different treatments for 20 days, quantified by RT-PCR. Data are expressed as means ± SD (N = 3 independent experiments). **i** CLSM images of mBMSCs demonstrating no distinct morphological change after co-incubation with M1 macrophages followed by treatment with ZMTP for 72 h. Rhodamine phalloidin and Hoechst were applied for stanning the skeleton and nuclei of cells, respectively. A representative image of three replicates from each group is shown. Scale bar, 100 μm. Source data are provided as a Source Data file.

used for the construction of an AIA model, which has been demonstrated to present similar pathologic features to human rheumatoid arthritis, such as chronic inflammation and bone destructions[43,44]. Such an animal model was constructed by injecting complete Freund's adjuvant directly at the right hind ankle joint of mice, then the immunization was let to continue for 15 days, which would lead to a distinct arthroncus of joint (Supplementary Fig. 23). In this study, the degree of joint swelling was obtained by calculating the ratio of the width of right inflamed ankle joints to the left normal ones ($W_R/W_L$). Twenty arthritic mice were divided into 4 experimental groups randomly and injected with PBS or PBS containing MnTCPP$^+$, Zn$^{2+}$ or ZMTP, at the pathological sites of arthritis. Normal mice without establishing AIA model were set as a control group for comparison. During the subsequent 21 days of observation period after different treatments, significant alleviation of ankle joint swelling was monitored in ZMTP group (Fig. 7a and Supplementary Fig. 24), while MnTCPP$^+$ and Zn$^{2+}$ alone presented negligible therapeutic effects, as they failed to show antioxidative and anti-inflammatory properties. Insignificant body weight changes of mice in the five groups have been recorded during the 15 days of immunization process (for arthritic mice) and 21 days of therapeutic process (Supplementary Fig. 25).

At the end of therapeutic process on day 36, all the mice were sacrificed and their right hind ankle joint tissues were obtained. Then the RNA was totally extracted and the expression levels of IL-6, IL-1β and ALP were measured by RT-PCR (Fig. 7b). The expressions of the two proinflammatory cytokines (IL-6 and IL-1β) are significantly downregulated after ZMTP treatment, while the level of ALP was upregulated in Zn$^{2+}$ and ZMTP groups, in consistence with the results of cellular experiments. The obtained joint tissues were also used for micro-CT scanning and histomorphometric analysis. Compared with normal mice, distinct erosion of ankle joints from arthritic mice can be visualized through the micro-CT images after 3D reconstruction (Fig. 7c). MnTCPP$^+$ and Zn$^{2+}$ alone failed to ameliorate the symptom significantly, while the ankle joints in ZMTP group were observed to show a normal trabecular structure and a smooth bone surface similar to those from healthy mice, suggesting the desired antiarthritic efficacy of the 2D MOF. Several key histomorphometric indexes based on the micro-CT scanning, such as bone volume/total volume (BV/TV), trabecular number (Tb.N) and separation (Tb.Sp), indicated a distinct recovery of bone microstructure of ankle joints after treatment with ZMTP nanosheet (Fig. 7d–f). Such a therapeutic efficacy of the 2D MOF may be attributed to the synergistic effect of anti-inflammation and pro-biomineralization by the material, which is not achievable by single MnTCPP$^+$ or Zn$^{2+}$ treatment due to the lack of potent antioxidative property. Moreover, safranin-fixed green staining further demonstrates cartilage erosion of ankle joints of arthritic mice (Fig. 7g). Significant therapeutic effect of ZMTP could also been demonstrated through the staining.

As discussed in the above sections, conventional antioxidants such as ascorbic acid, gallic acid and reduced glutathione, only serve as reactants rather than catalysts as the constructed ZMTP nanosheets function in this work, therefore they can only scavenge ROS stoichiometrically rather than catalytically. During the rheumatoid arthritis treatment, the single administration of ascorbic acid, gallic acid or reduced glutathione failed to elicit a continuous antiarthritic effect, finally the symptoms of arthritis would reappear (Supplementary Fig. 26 and 27). The value of $W_R/W_L$ at the end of therapeutic process on day 36 (i.e., $W_R/W_L(36)$) was used to compare the anti-inflammatory efficacies of different antioxidants with that of ZMTP nanosheet, while the high $W_R/W_L(36)$ values of the three non-catalytic antioxidants demonstrate their compromised anti-inflammatory efficacies

(Supplementary Fig. 26 and Supplementary Table 8). Additionally, the destroyed bone structures could not be repaired significantly by the three antioxidants (Supplementary Fig. 28–30). Therefore, the 2D MOF constructed in this study as an artificial antioxidase (antioxidant nanozyme) provides a novel nanocatalytic therapeutic approach that shows significant and long-lasting anti-inflammatory efficacy, while the Zn component further benefits bone regeneration. Such double therapeutic functions are hard to achieve by conventional non-catalytic antioxidants.

The comprehensive biosafety of ZMTP nanosheet has been further investigated in normal mice without establishing AIA model. Fifteen normal mice were divided into three experimental groups randomly ($N = 5$), followed by intraarticular injection with PBS or PBS containing varied concentrations of ZMTP. All key hematological, hepatic, and renal indicators of mice in different groups display insignificant changes (Supplementary Fig. 31 and Supplementary Table 9), indicating favorable biocompatibility of this 2D material. In addition, we also used normal mice to be intraarticularly injected with PBS or PBS containing MnTCPP$^+$, Zn$^{2+}$ or ZMTP, and the major organs of these mice (hearts, livers, spleens, lungs, and kidneys) after H&E staining show negligible pathological changes in different experimental groups (Supplementary Fig. 32), further suggesting that the 2D MOF is highly biocompatible for rheumatoid arthritis therapy.

## Discussion

In this work, a 2D MOF nanosheet catalyst has been constructed as an artificial antioxidase (antioxidant nanozyme) for nanocatalytic rheumatoid arthritis treatment. Such a catalyst was constructed via the coordination of a manganese porphyrin MnTCPP$^+$ and Zn$^{2+}$, between which the MnTCPP$^+$ provides an active Mn$^{III}$ site with a coordination environment analogous to those of the active metal sites of Mn-SOD and catalase, while Zn$^{2+}$ upregulates the redox potential of the coordinated MnTCPP$^+$, making the 2D MOF present considerable SOD- and catalase-like activities. The catalytic O$_2^{\bullet-}$ disproportionation by the material is based on the mechanism of outer-sphere proton-coupled one-electron transfer (the conversion between hydroxo-aquaMn(III) and monoaquaMn(II)), while the catalytic H$_2$O$_2$ disproportionation involves inner-sphere proton-coupled two-electron transfer (the conversion between hydroxo-aquaMn(III) and oxo-hydroxoMn(V)). This nanosheet is quite stable in solution in a short term of several hours but can degrade and release Zn$^{2+}$ during the subsequent long-term action by fluid stress. The cellular anti-inflammatory action of the nanosheets is achieved by scavenging intracellular ROS of M1 macrophages and promoting their transition to M2 phenotype, which largely prevents mBMSC apoptosis, while the Zn$^{2+}$ released from the degrading nanosheets upregulates the ALP level of mBMSCs, facilitating osteogenic differentiation and biomineralization. In vivo AIA model further evidences the favorable antiarthritic performance of the 2D MOF, indicating its high feasibility for rheumatoid arthritis therapy. It is expected that such a nanocatalytic antioxidation concept may be instructive to future multiple anti-inflammatory treatments.

## Methods

**Chemicals and reagents.** Zn(NO$_3$)$_2$·6H$_2$O and Na$_2$SO$_4$ were purchased from Sinopharm. Pyrazine was acquired from Alfa Aesar. PVP, Mn-SOD, catalase, complete Freund's adjuvant, and L-ascorbic acid were acquired from Sigma–Aldrich. MnTCPP$^+$ monochloride, TCPP$^+$ monochloride, FeTCPP$^+$ monochloride, gallic acid and reduced glutathione were provided by J&K Scientific. N,N-dimethylformamide (DMF) was bought from Shanghai Lingfeng Chemical Reagent Co., Ltd. CeO$_2$ nanoparticles were purchased from Jiangsu XFNANO. SOD assay kit, catalase assay kit, Hoechst, LPS, and alizarin red S staining kit were

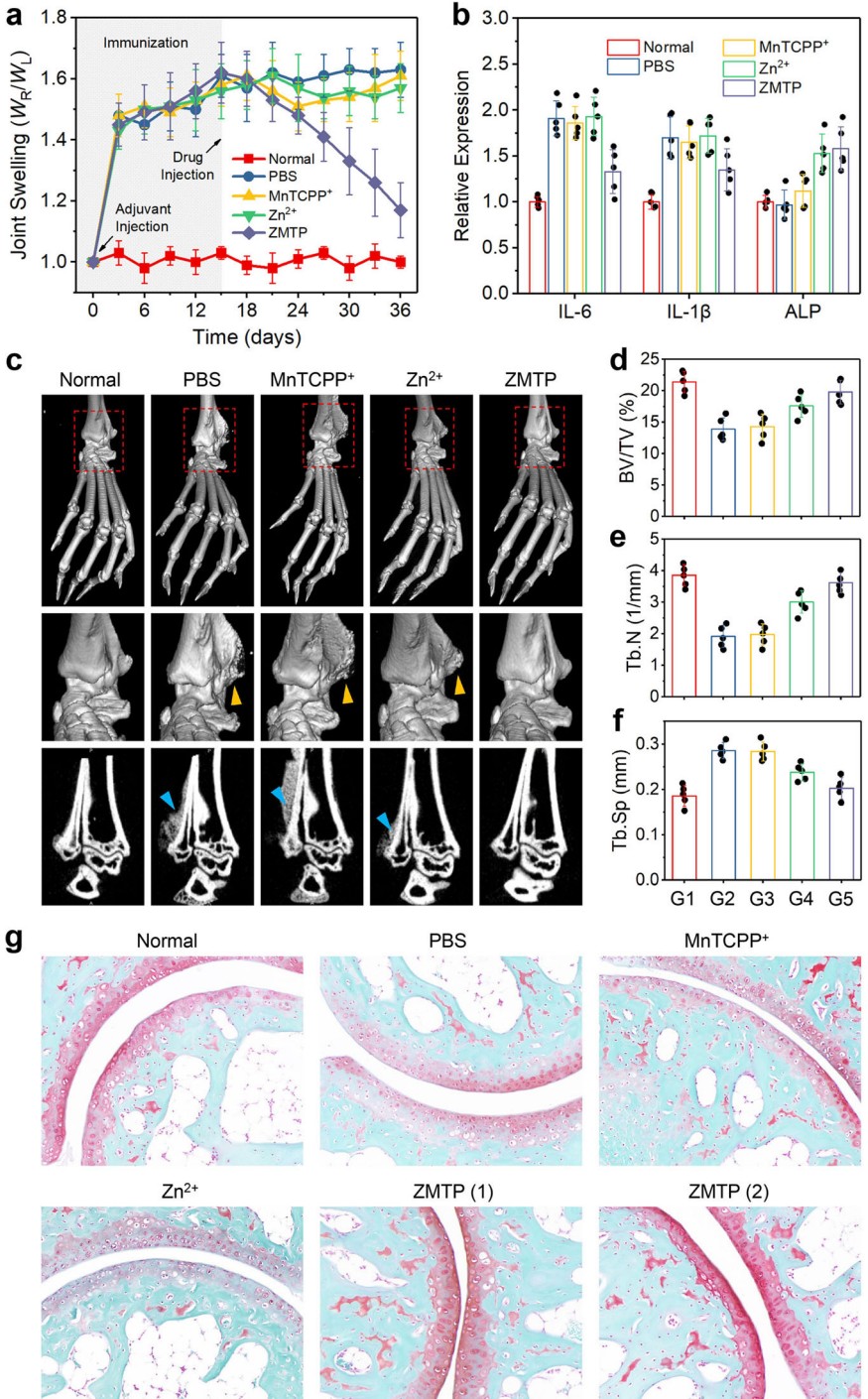

**Fig. 7 Antiarthritic function of ZMTP in vivo. a** Joint swelling behaviors of normal mice or arthritic mice after treatment with PBS or PBS containing MnTCPP[+], Zn[2+] or ZMTP. The degrees of right hind ankle joint swelling were determined by calculating the width ratio of the right ankle joints to the left ones ($W_R/W_L$). Data are expressed as means ± SD ($N = 5$ biologically independent animals). **b** Expressions of mRNAs of IL-6, IL-1β and ALP in joint tissues of mice in different groups on day 36. The mRNA expressions in joint tissues of normal mice were normalized as 1. Data are expressed as means ± SD ($N = 5$ biologically independent animals). **c** Micro-CT images of joints of mice in different groups on day 36 after 3D reconstruction. The micro-CT images of trabecular were also provided. A representative image of five replicates from each group is shown. Yellow and blue triangle marks reveal the abnormal bone structures due to inflammation. **d–f** Histomorphometric analysis such as BV/TV (**d**), Tb.N (**e**), and Tb.Sp (**f**) for joints of mice in different groups on day 36. G1 to G5 reveal normal, PBS, MnTCPP[+], Zn[2+] and ZMTP groups, respectively. Data are expressed as means ± SD ($N = 5$ biologically independent animals). **g** Safranin-fixed green staining of joints of mice in different groups on day 36. A representative image of five replicates from each experimental group is shown. Two samples from ZMTP group are shown. The degree of pink color indicates the relative quantity of chondrocytes. Scale bar, 100 μm. Source data are provided as a Source Data file.

acquired from Beyotime. Dulbecco's modified eagle's medium (DMEM), PBS, fetal bovine serum (FBS), and penicillin/streptomycin were obtained from YoBiBio. Annexin V-FITC, PI and CCK-8 assay were provided by 7seabiotech. Mesenchymal stem cell medium was bought from ScienCell. FITC, DCFH-DA and rhodamine phalloidin were provided by Solarbio. Paraformaldehyde (PFA) was obtained from Ribiology.

**Synthesis of ZMTP nanosheets**. Zn(NO₃)₂·6H₂O (3.0 mg), pyrazine (0.8 mg), PVP (20.0 mg), and MnTCPP⁺ monochloride (4.4 mg) were dispersed and homogenized in the mixture of DMF and ethanol (16 mL, volume ratio = 3:1). After sonication for 10 min, the solution was heated to 80 °C and allowed to reaction for 24 h. The resulting dark brown products were obtained by centrifugation at 8000 g for 2 min, then further washed with ethanol twice. To acquire the nanosheets with a small lateral size, the product was redispersed into ethanol and sonicated in ice bath with a sonication probe for 3 h (Power: 1000 W). The resulting mixture was further centrifuged at 20,000 g for 10 min to obtain the ZMTP nanosheets with a smaller lateral size. It is noted that the isostructural Zn-ZnTCPP-PVP nanosheets and Zn-FeTCPP-PVP nanosheets were prepared through the similar procedures by using equivalent TCPP⁺ monochloride and FeTCPP⁺ monochloride respectively instead of MnTCPP⁺ monochloride.

**Characterization**. TEM images, element mappings, SAED patterns, HAADF image, and EDS result were recorded on JEM-2100F electron microscope (JEOL). SEM image was obtained on SU9000 microscope (HITACHI). X-ray diffraction was conducted on Ultima IV (Rigaku). AFM measurement was conducted on NTEGRA (NT-MDT). XPS spectrum was obtained on ESCAlab250 spectrometer (Thermo Fisher Scientific). FTIR spectra measurement was performed on a Nicolet iS 10 spectrometer (Thermo Fisher Scientific). UV-Vis absorption spectra were obtained by UV-3600 (Shimadzu). CV and CA results were obtained on a CHI 700E workstation (CH Instruments). The concentration of released Zn element was determined by an ICP-OES (Agilent). Confocal fluorescence images were acquired on FV1000 (Olympus). Flow cytometric analysis was conducted on CytoFLEX (Beckman). RT-PCR was performed on 7900 HT (ABI). Micro-CT scanning images were acquired on Skyscan1176 (Bruker).

**Electrochemical response**. A standard three-electrode cell was used for recording CV and CA curves. A nickel foam electrode was used as the working electrode, an Ag/AgCl electrode was used as the reference electrode, a carbon rod electrode was used as the auxiliary electrode.

*CV*. CV curves were obtained within a voltage range between −1.0 and 0 V at a scan rate of 50 mV s⁻¹. A mild acidic NaCl aqueous solution (0.1 M, pH = 6.6) as the supporting electrolyte solution was added with MnCl₂ (1 mM) or MnTCPP⁺ monochloride ([Mn] = 1 mM) or ZMTP nanosheets ([Mn] = 1 mM) and purged with nitrogen for 10 min before CV measurements.

*CA*. Fresh electrolyte solution containing ZMTP nanosheets ([Mn] = 1 mM) was also used for CA measurement, which was subjected to the periodic actions of two step potentials (high potential: −0.5 V; low potential: −1 V; period: 120 s). Additionally, a new round of CA measurement was further conducted using a fresh electrolyte solution containing the same concentration of ZMTP nanosheets, but the step potentials were re-established (high potential: 0 V; low potential: −1 V; period: 120 s). During the two rounds of CA measurements, four aliquot solutions (1 mL) were extracted from the solution at the end of the former half period of the first period during the first round of CA measurement, the end of the first complete period of the first round of CA measurement, the end of the former half period of the first period during the second round of CA measurement, as well as the end of the first complete period of the second round of CA measurement, respectively, whose UV-Vis absorption spectra were further monitored.

**Antioxidase-mimicking activities**. The O₂•⁻ and H₂O₂ inhibition rates of ZMTP nanosheets were determined by using SOD assay kit and catalase assay kit at mild acidic solutions (pH = 6.6) following the manufacturer's protocols. The catalytic activities of single MnTCPP⁺ and Zn²⁺ were also measured respectively for comparison with that of ZMTP. Additionally, the O₂•⁻ and H₂O₂ inhibition rates of isostructural Zn-ZnTCPP-PVP and Zn-FeTCPP-PVP nanosheets were also evaluated. To investigate the stability of ZMTP nanosheets during catalysis, the nanosheets were added into the acidic NaCl aqueous solution (0.1 M, pH = 6.6) and subjected to periodic actions of two step potentials (high potential: 0 V; low potential: −1 V; period: 120 s). An aliquot (1 mL) was extracted from the solution after different cycles of voltage transitions, followed by O₂ supplementation for keeping the valence of Mn center to +3. Then the antioxidase-mimicking activities of ZMTP nanosheets in these aliquots were measured. The catalytic antioxidative efficiency of ZMTP nanosheets was further compared with natural enzymes (Mn-SOD and catalase) and typical nanocatalyst (CeO₂ nanoparticles), as well as several conventional molecular antioxidants (ascorbic acid, gallic acid, and reduced form of glutathione).

**Long-term degradability**. Mild acidic PBS (pH = 6.6) containing ZMTP nanosheets (500 ppm) was subjected to magnetic stirring for investigating the long-term degradability of the nanosheets under the action of differential fluid stress. At varied time points (0, 2, 6, 24, 48, 72, and 96 h), an aliquot solution (1 mL) was extracted from the mixture and centrifuged. The obtained supernatant was used for quantifying the concentration of released Zn element from the nanosheets by ICP-OES, while the resulting precipitate was further dispersed into ethanol for subsequent TEM observation.

**Cell culture**. Raw 264.7 cells (murine mononuclear macrophage cell line) were kindly provided by Cell Bank/Stem Cell Bank, Chinese Academy of Sciences. They were cultured in DMEM added with 10% FBS as well as 1% penicillin/streptomycin. The mBMSCs (murine bone mesenchymal stem cell line) were kindly provided by ATCC. They were cultured in mesenchymal stem cell medium supplemented with 5% FBS, 1% penicillin/streptomycin, and 1% mesenchymal stem cell growth supplement.

**Antioxidative action in macrophage**

*Cellular uptake*. Raw 264.7 cells were treated with LPS (10 ppm) for 2 days, aiming to promoting their polarization toward M1 phenotype. After that, the culture medium was discarded, and the cells were washed with PBS three times, which were further treated with fresh culture medium dispersing FITC-labeled ZMTP nanosheets (40 ppm). Hoechst was used for staining nuclei before observation under a confocal fluorescence microscope.

*Cytotoxicity*. M1 macrophages were treated with culture medium containing different concentrations of ZMTP nanosheets for 24 h or 48 h, then the cell viability was determined by CCK-8 assay.

*ROS scavenging*. M1 macrophages were treated with culture medium containing MnTCPP⁺, Zn²⁺ or ZMTP for 48 h, then the ROS probe DCFH-DA was used for cellular ROS imaging. Primary Raw 264.7 cells without LPS treatment were set as the control group. These cells were observed under confocal fluorescence microscope. Additionally, M1 macrophages treated with culture medium containing different concentrations of ZMTP nanosheets were also stained with DCFH-DA and analyzed by a flow cytometer. The results were obtained via CytExpert (version 2.2) and then processed via FlowJo (version 10.0).

*Phenotype reprograming*. M1 macrophages after different treatments were subjected to trypsinization and PBS washing, and the total RNA was extracted. The expressions of IL-6, IL-1β, TNF-α, Arg-1 and IL-10 were quantified by RT-PCR. The morphological changes of cells were also observed by optical microscope.

**Osteogenesis**

*Cell activity*. M1 macrophages and mBMSCs were incubated in a transwell system. After varied treatments, the cell viabilities of mBMSCs in different groups were determined by CCK-8 assay, while the apoptotic mechanism of cells was investigated by flow cytometry. For evaluating the direct effect of different treatments on the viabilities of mBMSCs, these cells were also directly seeded in 6-well plates followed by different treatments. Then CCK-8 assay was applied for determining cell viabilities.

*Osteogenic differentiation*. The effects of Zn²⁺ and ZMTP nanosheets on the biomineralization in the presence of mBMSCs were investigated by directly seeding mBMSCs in 6-well plates followed by indicated treatments. After co-incubation for 20 days, alizarin red S staining kit was applied for evaluating the generation of mineralized nodules. Additionally, M1 macrophages and mBMSCs were also seeded in the transwell system, followed different treatments for 20 days. The mRNAs of OPN, OCN, ALP, COL 1, and RUNX 2 were exacted from mBMSCs and quantified by RT-PCR. The morphological change of mBMSCs was also evaluated by stanning the cells with rhodamine phalloidin and Hoechst for imaging the cell skeleton and nuclei, respectively.

**Antiarthritic effect**. The animal-related experiment procedures in this work were conducted according to the guidelines approved by the Animal Ethics Committee of Shanghai Tenth People's Hospital, Tongji University School of Medicine (SHDSYY-2020-Z0026/1). Eighty female 8-week-old Balb/c mice were provided by Charles River, which were then housed under conditions of a light/dark cycle of 12 h, an ambient temperature of 25 ± 2 °C, and a humidity of 60 ± 10%. 25 of them were selected for evaluating the antiarthritic efficacy of ZMTP nanosheets, which were randomly divided into five groups (N = 5). The first group was set as control group without any treatment, while mice in the following four groups were injected with complete Freund's adjuvant (20 μL) in their right hind ankle joints, for establishing an AIA model. The left hind ankle joints of these mice were injected with PBS (designated as day 0). The widths of joints were recorded every three days, and the degree of joint swelling was indicated by calculating the ratio of the width of right inflamed ankle joints to the left normal ones ($W_R/W_L$). By using this method, the impact of natural joint growth on ultimate evaluation results was eliminated. After immunization for 15 days (day 15), the arthritic mice in the four AIA groups were intraarticularly injected with PBS (20 μL) or PBS containing

MnTCPP$^+$ monochloride (10 mg/kg), Zn(NO$_3$)$_2$ (10 mg/kg), or ZMTP nanosheets (10 mg/kg) at the pathological sites of arthritis.

**Histomorphometry**. On day 36, the mice in the five experimental groups were sacrificed and their right ankle joint tissues were obtained. Then micro-CT scanning was conducted for reconstructing the 3D structures of ankle joints as well as evaluating their histomorphometric characteristics. Several histomorphometric parameters, such as BV/TV, Tb.N, and Tb.Sp were quantified.

**Proinflammatory cytokine regulation**. The total RNA was extracted from the right ankle joint tissues for determining the expressions of mRNAs of IL-6, IL-1β and TNF-α by RT-PCR.

**Histology**. The obtained ankle joint tissues were decalcified over three weeks, then sectioned followed by safranin-fixed green staining.

**Efficacy comparison**. Fifteen normal Balb/c mice were subjected to complete Freund's adjuvant treatment (20 μL) in their right hind ankle joints for establishing the AIA model. Then these arthritic mice were divided into three groups randomly ($N = 5$) and injected with PBS (20 μL) containing ascorbic acid, gallic acid or reduced glutathione (10 mg/kg) at the pathological sites of arthritis. The value of $W_R/W_L$ at the end of therapeutic process on day 36 (i.e., $W_R/W_L(36)$) was used to compare the anti-inflammatory efficacies of different antioxidants with that of ZMTP nanosheet. The right ankle joint tissues of mice in the three groups were also harvested on day 36 for micro-CT scanning and safranin-fixed green staining.

**Biosafety**

*Hematology*. Fifteen normal Balb/c mice were divided into three groups randomly ($N = 5$), followed by intraarticular injection with PBS (20 μL) or PBS containing varied concentrations of ZMTP nanosheets (10 mg/kg, 20 mg/kg). On day 21 the blood samples (0.8 mL) were collected from these mice, and the serum biochemistry was evaluated.

*Histology*. Twenty normal Balb/c mice were divided into four groups randomly ($N = 5$), followed by intraarticular injection with PBS (20 μL) or PBS containing MnTCPP$^+$ monochloride (10 mg/kg), Zn(NO$_3$)$_2$ (10 mg/kg), or ZMTP nanosheets (10 mg/kg). On day 21 the major organs of mice, such as hearts, livers, spleens, lungs, and kidneys were obtained and fixed with PFA, followed by H&E staining.

**Statistical analysis**. Data were presented as mean ± standard deviation (SD) if $N \geq 3$.

**Reporting summary**. Further information on research design is available in the Nature Research Reporting Summary linked to this article.

## Data availability

The authors declare that all data needed to support the finding of this study are presented in the article and the Supplementary Information. This study uses publicly available data from the Protein Data Bank (PDB) under accession codes: 1N0J, 1DGF. Any data related to this work are available from the corresponding authors upon reasonable request. A reporting summary for this article is available as a Supplementary Information file. Source data is available for Figs. 2, 3, 4, 5, 6 and 7 and Supplementary Figs. 3b, 4, 10, 11, 13, 14, 15, 16, 17b, 25, 26, 29 and 31 in the associated Source Data file. Source data are provided with this paper.

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

## Acknowledgements
We greatly acknowledge the financial support from the National Natural Science Foundation of China (Grant No. 21835007), Key Research Program of Frontier Sciences, Chinese Academy of Sciences (Grant No. ZDBS-LY-SLH029), Basic Research Program of Shanghai Municipal Government (Grant No. 21JC1406000), and CAMS Innovation Fund for Medical Sciences (Grant No. 2021-I2M-5-012).

## Author contributions
B.Y. and J.S. designated the idea of this work. B.Y., H.Y., J.Y. and C.C. synthesized the 2D MOF and performed in vitro and in vivo experiments. B.Y. wrote the whole manuscript. J.S. supervised the project, revised the manuscript, and commented on it.

## Competing interests
The authors declare no competing interests.
