## [Peer Review File · Nature Communications]

Reviewers' Comments:

Reviewer #1:

Remarks to the Author:

The authors constructed a two-dimensional (2D) metal-organic framework (MOF) nanocatalyst as an artificial antioxidant for rheumatoid arthritis treatment. The 2D MOF characterized from structural and functional point of view and compared with the natural enzyme. The structure was evaluated for synthesis, size, stability, catalytic mechanism, antioxidative efficiency, long-term degradability in response to fluid stress, anti-inflammatory activity, osteogenic differentiation, in vivo antiarthritic function and rheumatoid arthritis treatment properties. The study is a new step in the field of nanozymes. The synthesized structure was comprehensively and completely evaluated. Based on the descriptions, the topic of this article is interesting and timely good align. The manuscript is well written, however there are some points that needs to be considered prior further consideration of the manuscript for publication.

1. The manuscript contains many complex and long sentences that needs to be carefully revised.
2. Since the term "nanozyme antioxidants" or "antioxidant nanozymes" is already an established and known terms, the authors may replace "antioxidase" with "antioxidant" for a better understanding of the paper's concept.
3. In the text, the size of these structures is often mentioned nanosheet, these materials are mostly in the category of superstructures (supramolecular structures), how are the dimensions of nano created? During synthesis? Or after synthesis and using various techniques such as ultrasonic, homogenizer and. . . . Has it become a nanosheet based materials? In the latter case, is the structure maintained or changed? Is there any data on the lack of restructuring?
4. All scale bars must be depicted in all figures, especially in Figure 2, Figure S12 Figure S17, and it is best to provide the image sources, such as XRD, TEM, and so on.
5. The most important data to determine the structure of these materials is X-ray crystallography (for molecular structure), While XRD (Fig S4) data shows a completely crystalline structure, which is empty in this article, Despite the large amount of data, this data is not provided, the reason should be explained.
6. Several typographical errors are present in the manuscript. It should be revised carefully. The acronyms can be used only if earlier presented by the full names. Also all the acronyms should be clarified. The entire manuscript should be typed with uniform line spacing. The punctuation marks aren't used correctly like faulty brackets, missed points etc. There are some wrong dictations like "dysfunction" that must be reviewed.
7. Please recheck the reference style in the manuscript because a number of references do not have the same style in the text. The references are relatively old. Please update this section and add some new references in the field. Many articles in the field of nanozymes have been published recently.

Reviewer #2:

Remarks to the Author:

In this manuscript, Yang et al. have reported the synthesis of a new type of 2D MOF and its application in nanocatalytic rheumatoid arthritis treatment. As the coordination environment of Mn(III) center of manganese porphyrin (MnTCPP+) is similar to those of the functional metal centers of natural SOD and catalase, the 2D MOF nanosheet can mimic the catalytic activities of the two antioxidant enzymes and catalyze sequential antioxidative reactions. Additionally, the Zn component can elevate the catalytic activities of manganese porphyrin as well as promote the process of biomineralization.

From the point view of academic research, this paper is well-written. The Introduction section provides a good story of structural to functional mimic of natural enzymes by using a nanomaterial for rheumatoid arthritis treatment, and this chemical concept may be instructive to future biomedical researches. The synergistic action between different components of the 2D nanocatalyst has also been demonstrated. In addition, the in vitro and in vivo experiments in this

work are very comprehensive, and the chemical mechanisms investigated in this study are deep. I am interested in the scholar presentation of Figure 3a and the equations in Table S2-S4.

Additionally, from the point view of practical application, the 2D MOF nanosheet is capable of achieving anti-inflammation and pro-biomineralization concurrently, while the catalytic characteristic of the nanosheet can reduce the administration dose required for efficient rheumatoid arthritis treatment, which is hard to achieve by conventional antioxidants as reactants (ascorbic acid, gallic acid and glutathione), as demonstrated in this manuscript. Therefore, the 2D MOF with intrinsic catalytic effect features a higher potential for future practical use.

Based on these considerations, I highly recommend this work to be published in Nature Communications after solving the following minor issues:

As we all know, in cancer therapy, drugs are usually administrated via intravenous injection. Comparatively, in this work the 2D MOF is administrated directly to the pathological site. For rheumatoid arthritis therapy, can the drugs be injected intravenously? Is there any innate biological mechanism for nanomedicines to be accumulated spontaneously in arthritic region (like EPR effect in tumor therapy)?

All the x-axis of the uv-vis spectra in this work (Figure 2j, Figure 4g, Figure S13, Figure S17b) should be wavelength rather than wavenumber, and the unit cm^{-1} should be corrected.

Some of the important information in the figure captions are missing. For example, in Figure 3b, 3f and 3g, the concentrations of materials/chemicals should be provided. In Figure 5e and 6h, how long are the incubation times?

In Figure 7g, further explanations on the safranin-fixed green staining of ankle joints should be provided for the better understanding by the readers.

For maintaining electric neutrality of the nanomaterial, Zn^{2+} and axial ligands are used to counteract the negative charges of benzoyloxys of MnTCPP and the positive charges of Mn(III) center, respectively. In Figure S9, the transition state of the 2D MOF should be negatively charged. Proper changes are suggested to be made.

Response to reviewer I.

Comments from reviewer I:

The authors constructed a two-dimensional (2D) metal-organic framework (MOF) nanocatalyst as an artificial antioxidant for rheumatoid arthritis treatment. The 2D MOF characterized from structural and functional point of view and compared with the natural enzyme. The structure was evaluated for synthesis, size, stability, catalytic mechanism, antioxidative efficiency, long-term degradability in response to fluid stress, anti-inflammatory activity, osteogenic differentiation, in vivo antiarthritic function and rheumatoid arthritis treatment properties. The study is a new step in the field of nanozymes. The synthesized structure was comprehensively and completely evaluated.

Based on the descriptions, the topic of this article is interesting and timely good align. The manuscript is well written, however there are some points that needs to be considered prior further consideration of the manuscript for publication.

Response: Thank you very much for the positive comment and kind recommendation. Please find the following detailed responses to your suggestions.

1. The manuscript contains many complex and long sentences that needs to be carefully revised.

Response: Thank you very much for the constructive suggestion. According to your suggestion, we have carefully checked the manuscript and the complex and long sentences have been rewritten.

2. Since the term “nanozyme antioxidants” or “antioxidant nanozymes” is already an established and known terms, the authors may replace “antioxidase” with “antioxidant” for a better understanding of the paper’s concept.

Response: Thank you very much for the kind suggestion. The term “artificial antioxidase” in the manuscript was used for better elucidation of the structural and catalytic characteristics of the constructed 2D MOF analogous to those of natural antioxidases. According to your suggestion, we instead strengthen the concept of “antioxidant nanozyme” in the revised manuscript by adding the term in the related sentences of the manuscript (Page 2, 5, 26 and 27).

3. In the text, the size of these structures is often mentioned nanosheet, these materials are mostly in the category of superstructures (supramolecular structures), how are the dimensions of nano created? During synthesis? Or after synthesis and using various techniques such as ultrasonic, homogenizer and. . . . Has it become a nanosheet based materials? In the latter case, is the structure maintained or changed? Is there any data on the lack of restructuring?

Response: Thank you very much for the kind question. As discussed on Page 8 of the manuscript, the solvothermal reaction resulted in the production of 2D MOF sheets with lateral sizes of over 3 μm , while probe sonication was followed to further reduce their lateral sizes to the nanoscale through an ultrasonic mechanical force.

The structure of the material has been well-maintained during probe sonication process, as the SAED pattern of the nanosheet (Figure 2f) shows a tetragonal structure identical to this category of pristine MOF (Adv. Mater. 2015, 27, 7372-7378). According to your question, the related discussion has been revised in the manuscript (Page 8 and 9).

4. All scale bars must be depicted in all figures, especially in Figure 2, Figure S12 Figure S17, and it is best to provide the image sources, such as XRD, TEM, and so on.

Response: Thank you very much for the kind suggestion. All scale bars have been provided in the figure captions, which is a common action for articles in Nature Communications. According to editorial request, we have further provided a Data availability section on Page 36 of the revised manuscript and clarified that all data related to this work are available from the corresponding authors upon reasonable request.

5. The most important data to determine the structure of these materials is X-ray crystallography (for molecular structure), While XRD (Fig S4) data shows a completely crystalline structure, which is empty in this article, Despite the large amount of data, this data is not provided, the reason should be explained.

Response: Thank you very much for the kind suggestion. In this work, the high-resolution TEM image and SAED pattern in the manuscript reveal the direct and visual evidences for the high crystallinity of the 2D MOF. Therefore, the XRD data were provided as the supplementary information for a supplementary elucidation.

6. Several typographical errors are present in the manuscript. It should be revised carefully. The acronyms can be used only if earlier presented by the full names. Also all the acronyms should be clarified. The entire manuscript should be typed with uniform line spacing. The punctuation marks aren't used correctly like faulty brackets, missed points etc. There are some wrong dictations like “dysfunction” that must be reviewed.

Response: Thank you very much for the constructive suggestion. According to your suggestion, we have checked the whole manuscript carefully and revised the typographical errors. The word “dysfunction” appears in the title of a reference (Ref 4) which cannot be further revised.

7. Please recheck the reference style in the manuscript because a number of references do not have the same style in the text. The references are relatively old. Please update this section and add some new references in the field. Many articles in the field of nanozymes have been published recently.

Response: Thank you very much for the constructive suggestion. Some of the literatures are books, and their reference formats are indeed different from research articles from scientific journals. According to your suggestions, several new references in the field of nanozymes have also been cited in the revised manuscript (Ref 34-36).

Response to reviewer II.

Comments from reviewer II:

In this manuscript, Yang et al. have reported the synthesis of a new type of 2D MOF and its application in nanocatalytic rheumatoid arthritis treatment. As the coordination environment of Mn(III) center of manganese porphyrin (MnTCPP+) is similar to those of the functional metal centers of natural SOD and catalase, the 2D MOF nanosheet can mimic the catalytic activities of the two antioxidant enzymes and catalyze sequential antioxidative reactions. Additionally, the Zn component can elevate the catalytic activities of manganese porphyrin as well as promote the process of biomineralization.

From the point view of academic research, this paper is well-written. The Introduction section provides a good story of structural to functional mimic of natural enzymes by using a nanomaterial for rheumatoid arthritis treatment, and this chemical concept may be instructive to future biomedical researches. The synergistic action between different components of the 2D nanocatalyst has also been demonstrated. In addition,

the in vitro and in vivo experiments in this work are very comprehensive, and the chemical mechanisms investigated in this study are deep. I am interested in the scholar presentation of Figure 3a and the equations in Table S2-S4.

Additionally, from the point view of practical application, the 2D MOF nanosheet is capable of achieving anti-inflammation and pro-biomineralization concurrently, while the catalytic characteristic of the nanosheet can reduce the administration dose required for efficient rheumatoid arthritis treatment, which is hard to achieve by conventional antioxidants as reactants (ascorbic acid, gallic acid and glutathione), as demonstrated in this manuscript. Therefore, the 2D MOF with intrinsic catalytic effect features a higher potential for future practical use.

Based on these considerations, I highly recommend this work to be published in Nature Communications after solving the following minor issues:

Response: Thank you very much for the comment and kind recommendation. Please find the following detailed responses to your suggestions.

1. As we all know, in cancer therapy, drugs are usually administrated via intravenous injection. Comparatively, in this work the 2D MOF is administrated directly to the pathological site. For rheumatoid arthritis therapy, can the drugs be injected intravenously? Is there any innate biological mechanism for nanomedicines to be accumulated spontaneously in arthritic region (like EPR effect in tumor therapy)?

Response: Thank you very much for the kind questions. Till now, there has been no widely-acknowledged biological mechanism on the spontaneous accumulation of nanoparticles/nanosheets in arthritic region in literatures, which leads to unknown accumulation efficiency at the pathological site and questionable therapeutic effects of nanomedicines by intravenous administration. Therefore, it is more appropriate to inject the antiarthritic drugs directly to the pathological regions of arthritis (which is well-defined and highly localized), according to the currently published reports: Nat. Nanotechnol. 2018, 13, 1182–1190; Angew.

Chem. Int. Ed. 2020, 59, 21864–21869; Angew. Chem. Int. Ed. 2021, 60,14458–14466.

2. All the x-axis of the uv-vis spectra in this work (Figure 2j, Figure 4g, Figure S13, Figure S17b) should be wavelength rather than wavenumber, and the unit cm^{-1} should be corrected.

Response: Thank you very much for the constructive suggestion. According to your suggestion, the x-axes of these Figures have been corrected.

3. Some of the important information in the figure captions are missing. For example, in Figure 3b, 3f and 3g, the concentrations of materials/chemicals should be provided. In Figure 5e and 6h, how long are the incubation times?

Response: Thank you very much for the constructive suggestion. According to your suggestion, these experimental details have been supplemented in the corresponding figure captions.

4. In Figure 7g, further explanations on the safranin-fixed green staining of ankle joints should be provided for the better understanding by the readers.

Response: Thank you very much for the constructive suggestion. According to your suggestion, the explanation has been supplemented in the caption of Figure 7g.

5. For maintaining electric neutrality of the nanomaterial, Zn^{2+} and axial ligands are used to counteract the negative charges of benzoyloxys of MnTCPP and the positive charges of Mn(III) center, respectively. In Figure S9, the transition state of the 2D MOF should be negatively charged. Proper changes are suggested to be

made.

Response: Thank you very much for pointing out this issue. The transition state of the 2D MOF in Figure S9 should be electroneutral. According to your suggestion, the hydroxo ligand has been corrected as an aqua ligand for maintaining electric neutrality of the nanomaterial.

Reviewers' Comments:

Reviewer #1:

Remarks to the Author:

Dear editorial board of Nature Communications

This is a very interesting article, the paper is well-written and the authors completely covered the comments, I strongly recommend that this work to be published in this journal.

Best

Reviewer #2:

Remarks to the Author:

I am satisfied with the revision and recommend this revised manuscript for publication as is.